# Dystrophin involvement in peripheral circadian SRF signalling

Corinne A Betts[1], Aarti Jagannath[2], Tirsa LE van Westering[1], Melissa Bowerman[1,3], Subhashis Banerjee[1], Jinhong Meng[4,7], Maria Sofia Falzarano[5], Lara Cravo[1], Graham McClorey[1], Katharina E Meijboom[1], Amarjit Bhomra[1], Wooi Fang Lim[1], Carlo Rinaldi[1,6], John R Counsell[4,7], Katarzyna Chwalenia[1], Elizabeth O'Donovan[8], Amer F Saleh[8,9], Michael J Gait[8], Jennifer E Morgan[4,7], Alessandra Ferlini[5], Russell G Foster[2], Matthew JA Wood[1,6]

**Absence of dystrophin, an essential sarcolemmal protein required for muscle contraction, leads to the devastating muscle-wasting disease Duchenne muscular dystrophy. Dystrophin has an actin-binding domain, which binds and stabilises filamentous-(F)-actin, an integral component of the RhoA-actin-serum-response-factor-(SRF) pathway. This pathway plays a crucial role in circadian signalling, whereby the suprachiasmatic nucleus (SCN) transmits cues to peripheral tissues, activating SRF and transcription of clock-target genes. Given dystrophin binds F-actin and disturbed SRF-signalling disrupts clock entrainment, we hypothesised dystrophin loss causes circadian deficits. We show for the first time alterations in the RhoA-actin-SRF-signalling pathway, in dystrophin-deficient myotubes and dystrophic mouse models. Specifically, we demonstrate reduced F/G-actin ratios, altered MRTF levels, dysregulated core-clock and downstream target-genes, and down-regulation of key circadian genes in muscle biopsies from Duchenne patients harbouring an array of mutations. Furthermore, we show dystrophin is absent in the SCN of dystrophic mice which display disrupted circadian locomotor behaviour, indicative of disrupted SCN signalling. Therefore, dystrophin is an important component of the RhoA-actin-SRF pathway and novel mediator of circadian signalling in peripheral tissues, loss of which leads to circadian dysregulation.**

## Introduction

Skeletal muscle is a dynamic structure in which myofilament turnover, maintenance and energy replenishment occur continually throughout the day [1]. Circadian transcriptomic studies in skeletal muscle indicate that ~3.4% of expressed skeletal muscle genes show rhythmicity, and that this differs between muscle types, specifically slow and fast muscle [2]. These rhythmic genes are involved in many central processes such as myogenesis, muscle lipid utilisation, protein metabolism and organisation of myofilaments, and very recently a key circadian gene, *Bmal1* (*Arntl*), was shown to be involved in impaired myogenicity in muscle of dystrophic mice [3].

Dystrophin is an integral sarcolemmal protein essential for muscle contraction and maintenance, absence of which leads to the devastating muscle wasting disease Duchenne muscular dystrophy (DMD) [4, 5]. Dystrophin has an actin-binding domain at the amino-terminus of the full-length isoform [6, 7, 8], which specifically binds and stabilises filamentous (F)-actin [9], an integral component of the RhoA-actin-serum response factor (SRF) pathway [10, 11, 12]. The RhoA-actin-SRF pathway is well described in muscle [13], and has since been shown to play an essential role in circadian signalling via systemic cues activating SRF in peripheral tissues [11]. Indeed SRF is a pivotal nuclear transcription factor, regulating more than 200 target genes [14] that are predominantly involved in cell-growth, migration, cytoskeletal organisation and myogenesis [15, 16], and one of the earliest SRF target genes to be identified was *Dmd* [13]. This integral relationship, combined with the understanding that the RhoA-actin-SRF pathway operates via a feed-back loop [17], intimates that the absence of dystrophin would have serious implications on SRF regulation. Indeed, in the reciprocal situation, in studies designed to mimic age related sarcopenia by disrupting skeletal muscle SRF expression, this resulted in atrophy, fibrosis, lipid accumulation and disturbed regeneration [18] which are all hallmarks of the DMD phenotype [19, 20, 21] further supporting their cyclical nature and mutual dependence.

Gerber's revolutionary work on the circadian regulation of the RhoA-actin-SRF pathway, eloquently describes how the hypothalamic suprachiasmatic nucleus (SCN), transmits systemic cues

[1]Department of Paediatrics, University of Oxford, South Parks Road, Oxford, UK   [2]Sleep and Circadian Neuroscience Institute (SCNi), Nuffield Department of Clinical Neurosciences, Oxford Molecular Pathology Institute, Dunn School of Pathology, University of Oxford, Oxford, UK   [3]School of Medicine, Keele University, Staffordshire, Wolfson Centre for Inherited Neuromuscular Disease, The Robert Jones and Agnes Hunt Orthopaedic Hospital, Oswestry, UK   [4]Dubowitz Neuromuscular Centre, Molecular Neurosciences Section, Developmental Neuroscience Programme, University College London Great Ormond Street Institute of Child Health, London, UK   [5]Department of Medical Sciences, Unit of Medical Genetics, University of Ferrara, Ferrara, Italy   [6]Muscular Dystrophy UK Oxford Neuromuscular Centre, University of Oxford, Oxford, UK   [7]National Institute for Health Research Great Ormond Street Hospital Biomedical Research Centre, London, UK   [8]Medical Research Council, Laboratory of Molecular Biology, Francis Crick Avenue, Cambridge, UK   [9]Functional and Mechanistic Safety, Clinical Pharmacology and Safety Sciences, R&D, AstraZeneca, Cambridge, UK

Correspondence: corinne.betts@paediatrics.ox.ac.uk

thereby activating RhoA in peripheral tissues (11). They demonstrate that diurnal polymerisation (F-actin) and de-polymerisation (globular (G)-actin) of actin influences SRF expression and transcription of specific downstream circadian targets (*Per1* and *Per2*) and output genes (*Nr1d1* and *Rora1*) (11, 12). However, disruption of this pathway by removing alternative upstream proteins intrinsic to the cascade has not been shown. Given the proximity of dystrophin to F-actin (which it specifically binds and stabilises) combined with its integral relationship with SRF transcription, we hypothesise that dystrophin loss leads to a shift in actin de-polymerisation, which affects SRF expression and downstream circadian gene expression, thus resulting in circadian dysregulation in DMD models.

## Results

To assess whether dystrophin loss disrupts the RhoA-actin-SRF pathway, an in vitro assay was designed using siRNAs targeting the *Dmd* gene to down-regulate dystrophin protein in a skeletal muscle cell line (H2K 2B4 myoblasts) (22). Differentiated myotubes were transfected twice with 100 nM siRNA, and collected 49 h after the second transfection, thereby representing circadian time 1 (CT1). Efficient transfection resulted in undetectable amounts of dystrophin protein (Fig 1A) and markedly reduced *Dmd* transcript levels (Fig 1B). Dystrophic conditions resulted in significantly altered RhoA activation (Fig 1C) and decreased F/G-actin ratios (Fig 1D). Additionally, there appears to be greater cytoplasmic MRTF accumulation under dystrophic conditions (not significantly different; Fig 1E). To illustrate circadian oscillation patterns of core clock genes involved in the transcriptional auto-regulatory feedback loop after abolition of dystrophin, cells were collected every 4 h over a 24-h period, and indicate significant alterations (Fig 1F). *Per1*, *Per2*, and *Clock* expression were significantly down-regulated at certain time points, whereas *Cry2* and *Arntl1* were up-regulated. Expression of other downstream SRF target genes in the actin-cascade, *Nr1d1* and *Acta* (12), was also significantly lower in dystrophin deficient samples, and interestingly *Srf* expression was significantly up-regulated. Although *Srf* up-regulation was unexpected, this may be due to activation of alternative compensatory pathways such as the ternary complex factor family of Ets domain proteins (MAPK pathway) which regulates transcription of growth responsive genes (12). Importantly, *Srf* up-regulation does not denote SRF activation via the RhoA-actin pathway, which would require nuclear translocation of MRTF. In order to investigate to what extent dystrophin loss in human DMD patients resulted in similar biochemical events, we obtained DMD muscle biopsies from patients (tissues collected between 8 and 10 AM in the morning and patients fasted from midnight the evening before biopsy). Interestingly, a wide array of mutations showed down-regulation of key RhoA-actin-SRF targets *PER1*, *PER2*, and *NR1D1* in nearly all samples (Fig 1G).

Given these in vitro and biopsy results, we predicted that SRF signalling in peripheral muscle would also be interrupted in dystrophic mice, leading to molecular deficits. The dystrophin–utrophin knockout (dKO) model presents with a severe phenotype that closely recapitulates disease in patients, specifically severe progressive muscular dystrophy, premature death and a plenitude

of physiological and molecular aberrations (23). The oscillation patterns of core clock genes in the *tibialis anterior* (TA) muscle of 5-wk-old male mice were assessed over a 24-h period (double plotted). This model revealed profound alterations in core clock gene expression, with markedly similar patterns to that observed in the dystrophic- H2K 2B4 myotubes. The amplitude of SRF-target genes *Per1*, *Per2*, and *Nr1d1* was lower in 5-wk-old dKO animals compared with C57BL10 (Fig 2A), suggesting dampened rhythmicity. Gene expression of *Cry1*, *Cry2*, *Arntl*, *Clock*, and other downstream targets (*Acta* and *Rora1*) were all significantly up-regulated at various ZTs (Zeitgebers). As observed in the H2K 2B4 model, *Srf* expression was again up-regulated under dystrophic conditions. F/G-actin ratios were significantly lower in 5-wk-old dKO animals compared with control animals (ZT1; Fig 2B) and littermate *mdx* animals (Fig S1), indicating a more profound impact on F-actin in dKO mice compared with the less effected *mdx* model. In dKO animals, the levels of nuclear MRTF protein trends lower (Fig 2C, ZT1), and MRTF cytoplasmic protein levels were significantly lower (ZT13) compared with control animals, which together suggests there is less total MRTF in dKO animals. It is important to note that F/G actin ratios exhibited diurnal changes in skeletal muscle of healthy mice (Fig 2B), and MRTF cytoplasmic fraction levels appear to oscillate also. These results, in combination with the in vitro data, intimates the actin pathway is indeed perturbed in muscle due to dystrophin loss, and components within the cascade display diurnal alterations.

We further anticipated that systemic cues from the central clock (SCN) to SRF in peripheral muscle would be interrupted. Although the dKO mouse model closely recapitulates the dystrophic phenotype in patients and is a good molecular model for DMD, its severe phenotype, including reduced lifespan (~5-8 wk) and marked reduction in activity, precludes extensive locomotor behaviour studies. As such, the less affected dystrophic model, *mdx*, was used for this extensive battery of locomotor tests. We show for the first time that dystrophin protein is expressed in the SCN of C57BL10 mice but not *mdx* animals (Fig 3A), and therefore it was pertinent to assess whether there were any obvious abnormalities in the circadian locomotor behaviour in dystrophic mice. Wheel-running activity of 20 wk old (symptomatic) male *mdx* and C57BL10 mice were recorded under various conditions, and representative actograms under 12 h:12 h light–dark (LD), 24 h dark (DD), and 24 h light (LL; Fig 3B) are shown. Under normal LD conditions, no significant differences in the number of bouts or total activity between mouse cohorts were observed, indicating that the endurance level of *mdx* mice was normal (Fig 3C). This observation was valuable for the interpretation of subsequent data, as it indicated differences between genotypes was not due to the muscle wasting phenotype of *mdx* mice, but rather signalling cues from the SCN. Interestingly, during the light phase of LD, activity of *mdx* mice was markedly reduced, and they exhibited delayed onset into dark phase (phase angle). After 6-h phase advance bouts, *mdx* mice were capable of re-entraining to the shifted cycle in a similar manner to control animals (Fig 3D). Animals were then placed in DD, where *mdx* animals again indicated a delayed onset (phase angle) on release into dark (Fig 3E), suggesting that their endogenous clock may be out of phase. Again endurance of *mdx* animals was maintained during DD and they ran for a similar period compared with C57BL10 animals; however, their free running period was significantly shorter

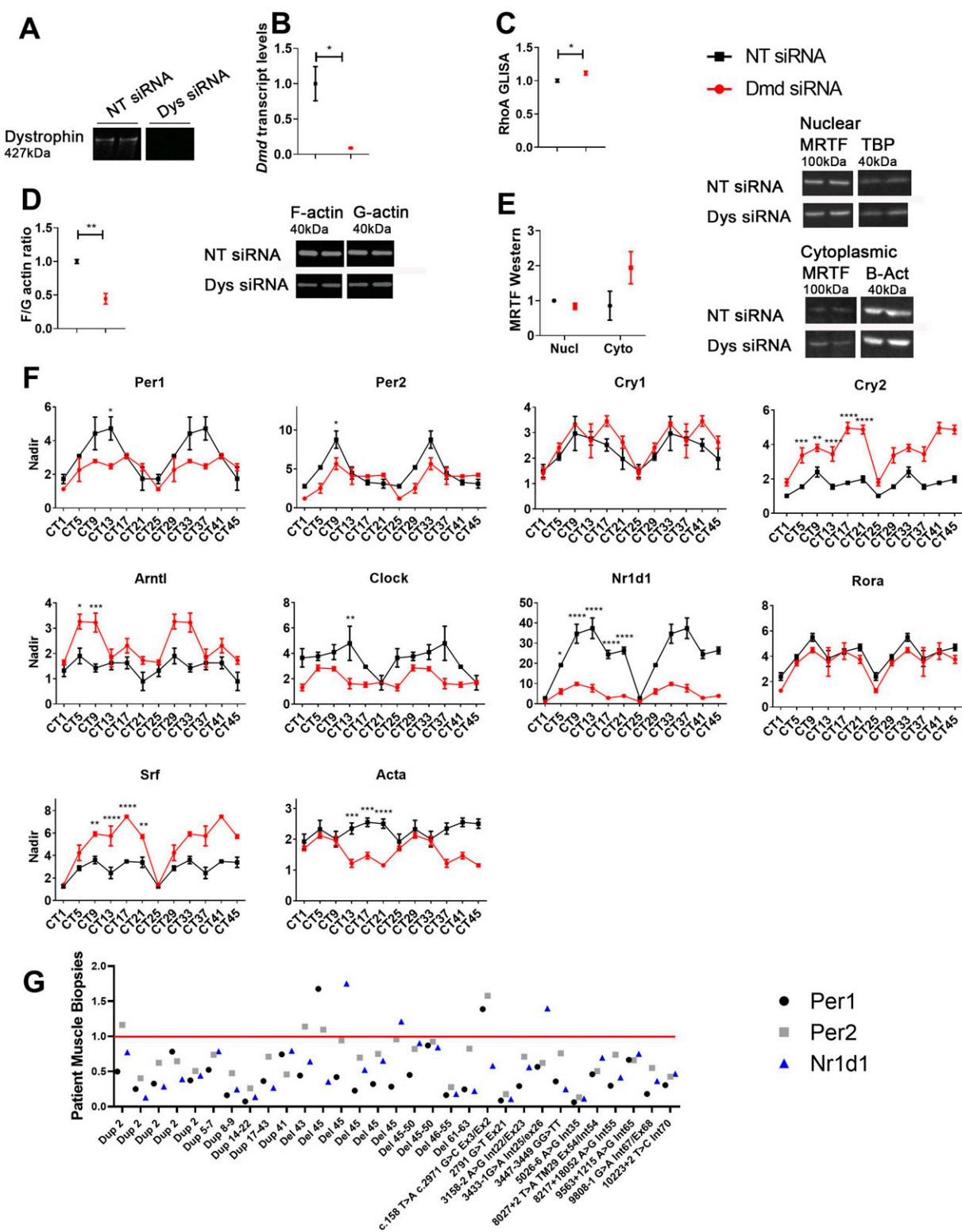

**Figure 1. Disruption of RhoA-actin-serum response factor (SRF) pathway in dystrophic cells and patient muscle biopsies.**
To mimic dystrophic conditions, H2K 2B4 myotubes were transfected with siRNAs targeting the *Dmd* gene (Dmd siRNA, red), and a non-targeting (NT siRNA, black) siRNA was used for control. **(A, B)** Dystrophin was successfully down-regulated at (A) protein and (B) the transcript level. **(C, D, E)** In addition, the absence of dystrophin resulted in (C) altered RhoA activity, (D) reduced F/G-actin levels, and (E) greater cytoplasmic MRTF accumulation (loading controls: nuclear TBP, and cytoplasmic β-actin). For gene transcript studies, cells were placed is serum free media and collected over a 24-h time course (double plotted-48 h-to show oscillation pattern). **(F)** Circadian time (CT) gene expression data indicates diurnal oscillation patterns and down-regulation of pertinent core clock genes (*Per1* and *Per2*) and down-stream targets for RhoA-actin-

(Fig 3F; *mdx* run half an hour shorter). During the DD phase, mice received a light pulse 4-h after they started exercise (CT16), and *mdx* mice displayed no difference in the ability to shift the clocks phase in response to this nocturnal light compared with C57BL10 animals (Fig 3G). In constant light, activity counts for *mdx* animals dropped dramatically, and free running period was also significantly reduced (Fig 3H). When considering our altered dystrophin-associated-RhoA-actin-SRF cascade hypothesis, altered activity may be due to dystrophin loss in the SCN leading to alterations in core clock gene, *Per2*, which has been associated with shorter circadian period and loss of circadian rhythmicity in constant darkness (24). The delayed phase angle in LD and DD, combined with extreme lack of activity during LL suggest a severe aversion to light stimuli in *mdx* animals. Long-term exposure to light has been shown to affect neurons in the SCN and reduce rhythmicity (25), which aggravated by the loss of dystrophin in the SCN, may explain the considerable changes in activity of *mdx* mice. Altogether, these data demonstrate the SCN is profoundly affected in dystrophic mice.

Male *mdx* and C57BL10 mice were also subject to repeated bouts of chronic jetlag (JL), whereby lighting conditions were advanced 6 h every week for 5 wk (mice weighed weekly). Age-matched *mdx* were significantly heavier than C57BL10 mice (15 wk old at start of protocol, 22 wk by end); therefore, a "weight matched" group (12 wk at start, 20 wk at end) was also assessed. "Weight-matched" *mdx* mice significantly increased in weight over 5 wk, whereas there was no increase in weight in the C57BL10 cohort (Fig 3I). The JL cohorts were further compared with non-jetlagged (NJL) groups. No significant differences were observed between JL and NJL C57BL10 cohorts (all mice 22 wk of age). Although NJL *mdx* animals were significantly heavier than both C57BL10 cohorts, most importantly the JL *mdx* cohort was significantly heavier than all other groups. Jetlag causes changes in phase of entrainment in the SCN and peripheral clock (26). Some clocks entrain faster than others which can cause internal desynchrony. If the synchronising signal such as SRF is lost (in this case due to disruption of RhoA-actin-SRF cascade), it is anticipated that there will be a greater disruption and desynchrony from jetlag protocols leading to changes in weight. This would account for why the C57BL10 animals resisted weight gain under the short JL conditioning protocol, whereas the *mdx* animals do. Together, these data indicate altered signalling between the SCN and peripheral tissues in dystrophin-deficient *mdx* mice.

As locomotor experiments were performed in the *mdx* model, gene expression patterns of core clock genes in the TA muscle of 20 wk old male mice were assessed over a 24-h period (double plotted). A shift in phase was observed for *Per1*, *Per2*, *Cry1*, *Cry2* and *Arntl1*, resulting in significant differences in gene expression at certain time-points within the day (Fig 4). For instance, the expression of *Per2* in *mdx* mice peaked at ZT5, but in C57L10 animals peaked at ZT9. Gene expression of downstream SRF target, *Rora1*, was significantly down-regulated (ZT1, ZT17, and ZT21), and minor alterations in *Nr1d1* (ZT21) and *Acta1* (ZT1) were observed.

Most importantly we observe alterations in integral components of the RhoA-actin-SRF cascade (in particular SRF targets and F/G-actin ratios) in all dystrophic models described. Although the *mdx* gene dataset differs to the dKO and H2K 2B4 myotube models, we regard the dKO model with greater esteem given its phenotype and correlation with patient disease progression (23), which is supported by the biopsy data (Fig 1H). It further correlates closely with cell-culture data in which dystrophin is specifically ablated (Fig 1F). However, to illustrate locomotive aberrations, and systemic cues with the SCN, it was imperative we look in the milder *mdx* model. The remarkable lifespan and generally mild phenotype of *mdx* mice is poorly understood, but it is likely due to multiple compensatory events that ensue and may account for the variances observed between the dystrophic models. As multiple inputs can regulate the clock, it is difficult to predict how the clock will react when one input is removed. In vivo complexities in the form of protein or signalling interactions with serum, hormones (particularly glucocorticoids as *Per2* has a glucocorticoid receptor-binding site) and neurological signals or mechanisms triggered to compensate for disturbed RhoA-actin-SRF pathway may be involved. One such compensatory mechanism is up-regulation of utrophin, a homologue of dystrophin, which is knocked out in dKO but present in *mdx* mice. Utrophin has been shown to bind and maintain F-actin polymerisation and therefore seems an obvious candidate in the RhoA-actin-SRF signalling cascade (27). We show the presence of utrophin gene and protein expression in *mdx* mice (Fig S2A); however, in H2K 2B4 myotubes, *Utrn* gene expression and protein levels were reduced (Fig S2B). Similarly, in DMD patient biopsies, *UTRN* gene expression was down-regulated in all samples with the exception of one, which incidentally also showed up-regulation of *PER1* and *PER2* for that sample (Fig S2C). Indeed, linear regression analysis of *UTRN* expression versus *PER1* and *PER2* indicate a significant correlation pattern (*PER1* R = 0.1687, *PER2* R = 0.4542; Fig S2D). Thus, utrophin may compensate for dystrophin loss in *mdx* animals, and loss of both utrophin and dystrophin may lead to greater F-actin instability and downstream SRF activity, as observed in the H2K 2B4 and dKO data.

To assess the RhoA-actin SRF cascade further, and determine whether ablation of other upstream components of the pathway, such as actin, or indeed SRF itself, results in similar changes in the expression of target genes, siRNAs were used to specifically knock-down *Srf* and *Acta* in H2K 2B4 myoblasts. To compare with dystrophin knock-down experiments, differentiated myotubes were transfected twice with 100 nM *Srf* and *Acta* siRNAs, alongside *Dmd* transfected myotubes, and collected 49 h after the second transfection, thereby representing CT1. Myotubes were also treated

---

SRF (*Nr1d1* and *Acta1*; housekeeping gene—*Gapdh*). **(G)** Muscle biopsies were obtained from an array of patients with different mutations or deletions in the dystrophin genes, and indicate down-regulation of RhoA-actin-SRF target genes in most cases (housekeeping genes—*RPL13a*). All samples were normalised to a pooled skeletal muscle sample from two healthy volunteers, as indicated by red line. For RhoA GLISA, F/G actin ratio and MRTF Westerns, data were normalised to NT control (n = 3; two-tailed *t* test). For RT-qPCR CT data, the nadir was determined as the minimum value across both treatments (Dmd and NT siRNAs) and CTs and applied to all samples; nadir normalised to 1 (n = 2–3 for each siRNA and time point; two-way ANOVA with Bonferroni post hoc test performed). Mean values reported with SEM; ***P < 0.001, **P < 0.01, *P < 0.05. For uncropped Western blot images and loading controls, see Source Data Fig 1. Source data are available for this figure.

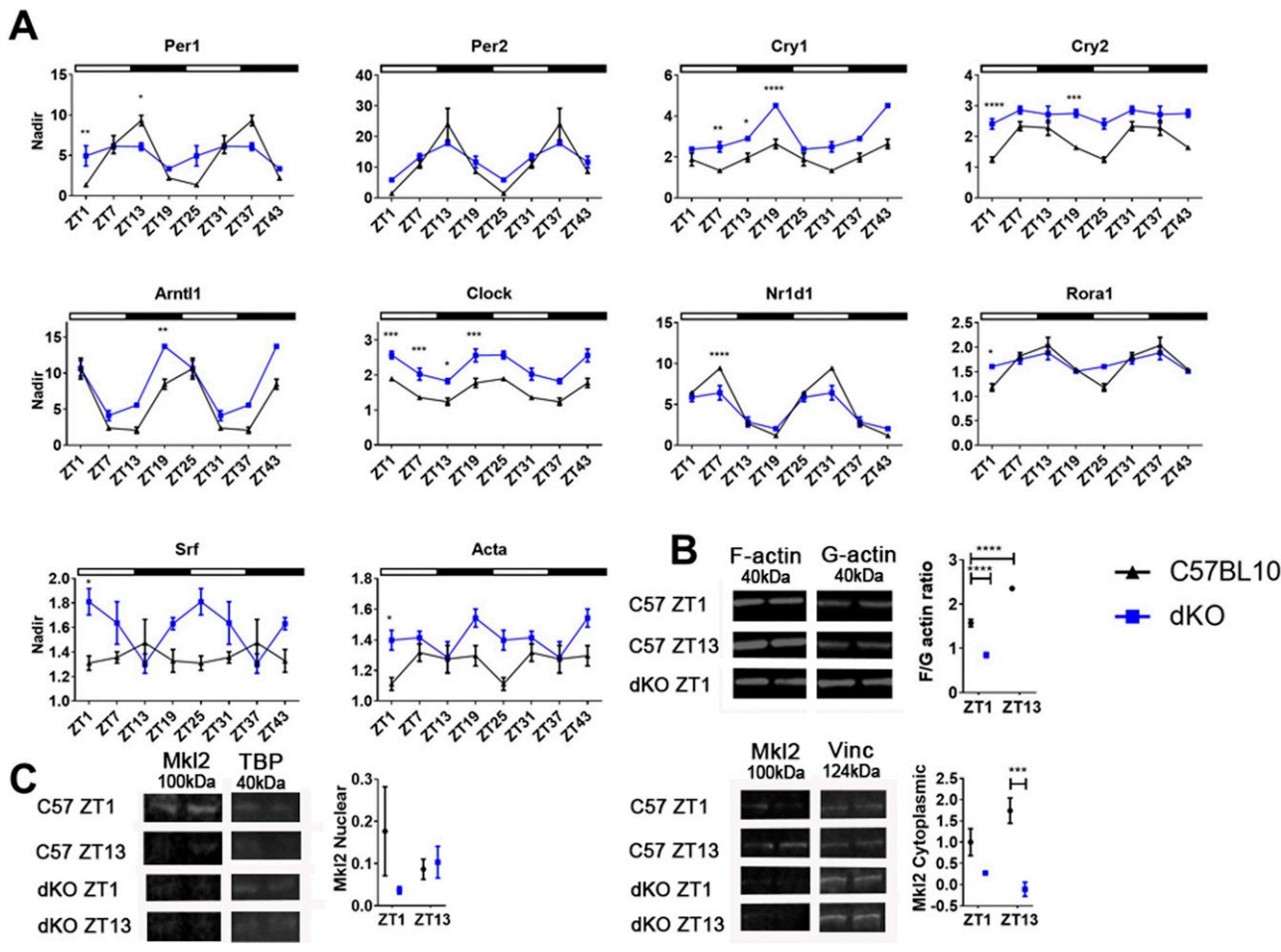

**Figure 2. Diurnal changes in RhoA-MRTF-serum response factor (SRF) cascade components in tibialis anterior of dystrophin-utrophin (dKO) mouse model.**
**(A)** Tissues were collected over a 24-h time course and double plotted (48 h) to better illustrate the (A) oscillation pattern of core clock genes in *tibialis anterior* (TA) of 5-wk-old *dKO* animals which were significantly altered compared to C57BL10 (Zeitgeber- ZT). Down-stream targets for RhoA-actin-SRF pathway (*Nr1d1*, *Rora1*, and *Acta*) and indeed *Srf* were also altered (*Gapdh* used as housekeeping gene). **(B, C)** F/G-actin protein ratios in *dKO* mice were significantly down-regulated, (C) as were cytoplasmic MRTF fraction levels (loading control: nuclear TBP, and cytoplasmic vinculin). In addition, C57BL10 animals exhibit diurnal changes in F/G actin ratio and MRTF fraction levels. Light and dark periods represented by outlined (light) and solid bars (dark). For RT-qPCR ZT data, the nadir was determined as the minimum value across both genotypes and ZTs and applied to all samples; nadir normalised to 1 (n = 3–4 for each for each genotype and time-point). For F/G actin ratio and MRTF westerns, data were normalised to C57BL10 (n = 3–4). One or two-way ANOVA with Bonferroni post hoc test performed. Mean values reported with SEM; ***$P < 0.001$, **$P < 0.01$, *$P < 0.05$. For uncropped Western blot images and loading controls, see Source Data Fig 2.
Source data are available for this figure.

with lower concentrations of siRNA to confirm gene expression was stable and that myotubes were healthy (Fig S3). Interestingly, *Srf*, *Acta* and *Dmd* down-regulation, appear to reciprocally modulate each other resulting in lower expression of all genes for all cohorts, that is, *Srf* knock-down results in lower *Dmd* and *Acta* gene expression and vice versa (Fig 5). In addition, all siRNA treatment groups resulted in reductions of RhoA-actin-SRF target genes, *Per1* and *Per2*, and *Nr1d1* and *Rora*. Together, this illustrates how intertwined and mutually dependent these genes are in maintaining homeostatic balance of the RhoA-actin-SRF cascade.

## Discussion

Because of the severe repercussions of most genetic disorders, many other symptoms of disease are often overlooked, as efforts are primarily aimed at targeting the underlying genetic defect. This is most certainly the case for DMD, a monogenic disorder resulting in muscle wasting and cardiomyopathy in affected boys (28, 29). These boys also experience abnormalities in sleeping patterns (30), and nocturnal hypoxaemia and hypercapnia (31) which may be attributed to the dystrophic phenotype-specifically the deterioration in respiratory muscles. Here we propose this may also be due to a circadian deficit.

We show circadian perturbations in a number of dystrophic models and suggest a mechanistic rationale for these changes by investigating components of the RhoA-actin-SRF cascade. This seemed a logical approach given F-actins interaction with dystrophin, and its importance in the RhoA-actin-SRF cascade, which is crucial for muscle homeostasis (11, 13). In the case of healthy muscle, RhoA regulates polymerisation (F-actin) and de-polymerisation (G-actin; Fig 6). During de-polymerisation G-actin preferentially binds to MRTF, but

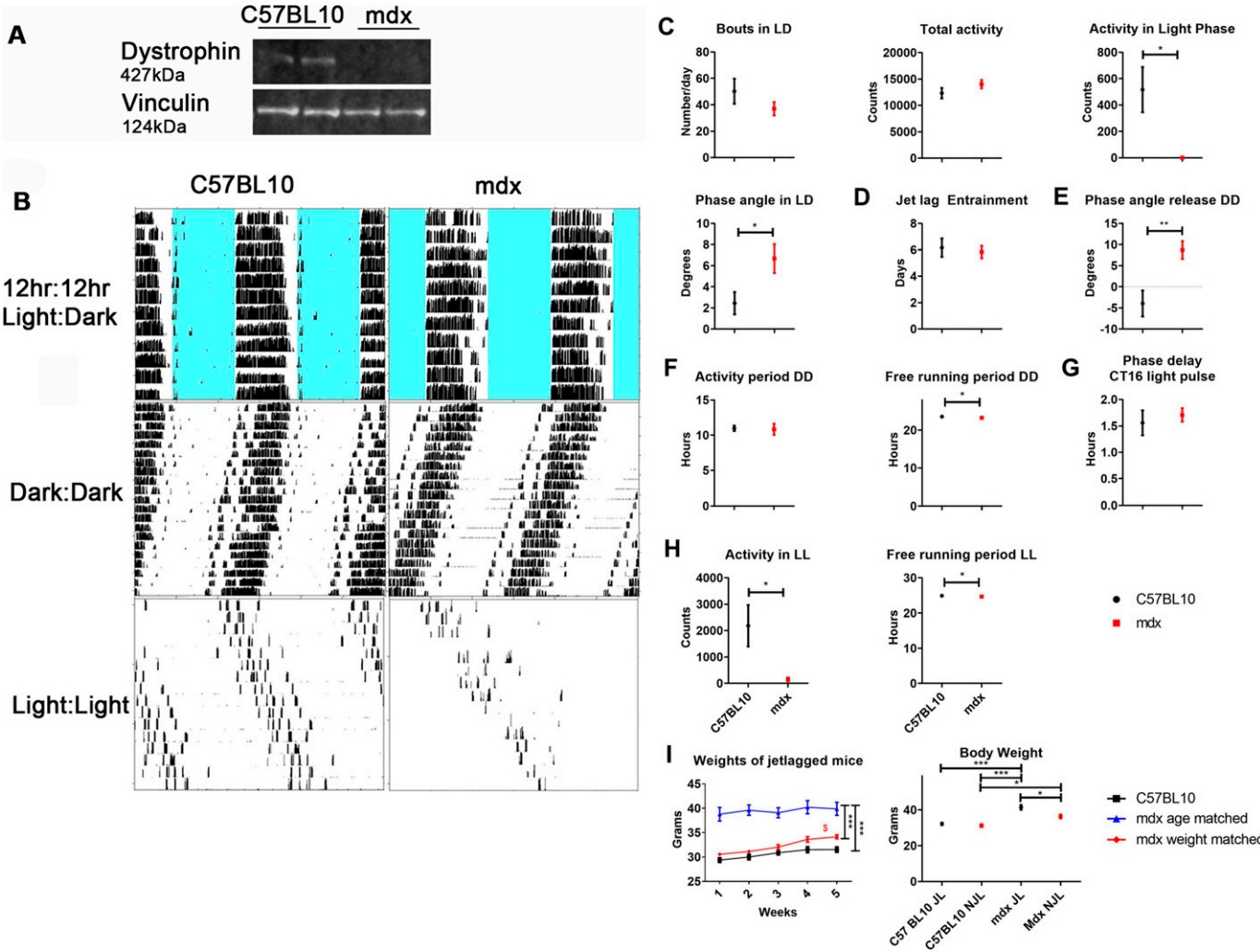

**Figure 3. Perturbed circadian rest-activity and response to jet-lag in dystrophic mouse model.**
**(A)** Dystrophin Western blot shows the absence of protein in the SCN of *mdx* animals (vinculin used as loading control). **(B)** Locomotor behaviour of *mdx* and C57BL10 mice were assessed and representative wheel-running activity blots (B) shown; 12 h:12 h light:dark cycle (LD), dark:dark (DD), and light:light (LL). **(C, D, E, F, G, H)** Data analysed and represented as graphs: (C) behaviour during LD, (D) after 6-h phase advancements, (E) phase angle on release into DD, (F) behaviour during DD, (G) phase delay after a light pulse 4-h post exercise commencement, (H) and behaviour during LL. **(I)** Weights of mice undergoing 5-wk jetlag protocol assessed (C57BL10, *mdx* age matched and *mdx* weight matched), as well as total body weight of jetlagged (JL) and non-jetlagged (NJL) *mdx* and C57BL10 cohorts. *t* test for comparisons between two groups (two-tailed). For weights recorded over the 5 wk of jetlag: black asterisk on right-one-way ANOVA comparing all three cohorts; dollar sign-one way ANOVA with Bonferroni post hoc between each cohort at each time point. For body weights with JL and NJL, one-way ANOVA with Bonferroni post hoc test performed (For wheel running activity $n = 5–6$, jetlag study $n = 3–4$; ***$P < 0.001$, **$P < 0.01$, *$P < 0.05$). For uncropped dystrophin Western blot and loading control, see Source Data Fig 3. Source data are available for this figure.

when this shifts to the polymerisation phase the G-actin pool diminishes and unbound MRTF translocates to the nucleus and influences SRF expression. SRF activation in turn regulates transcription of target genes—*Per1*, *Per2*, *Rora1*, *Nr1d1*, and *Acta*. Here, we propose that the absence of dystrophin reduces F-actin levels resulting in shifts to the de-polymerised state. G-actin binds to MRTF, which we show remains cytosolic thereby hampering SRF activation (Fig 6). In addition, we show for the first time that dystrophin is absent in the SCNs of dystrophic mice, and that these animals exhibit behavioural alterations, indicative of disrupted central circadian signalling within the SCN. As such, the lack of dystrophin results in circadian disruption that manifests with physiological and molecular alterations in dystrophic models.

Given that dystrophin regulates circadian signalling in peripheral tissues, this suggests that related dystrophin glycoprotein complex (DGC) proteins may be implicated in dystrophin and F-actin tethering, loss of which could also result in circadian dysregulation. Indeed this is supported by knock-down of *Srf* and *Acta*, alternative components of the RhoA-actin-SRF pathway, which also lead to down-regulation of target genes. Thus, muscular dystrophy disorders, such as limb-girdle muscular dystrophy (32), in which DGC and sarcomeric proteins are affected should be assessed for circadian abnormalities.

It is possible that some pharmacological interventions used to improve the muscle phenotype in DMD patients, may have inadvertently modulated circadian rhythm resulting in physiological improvements, such as melatonin (33) and glucocorticoids (34).

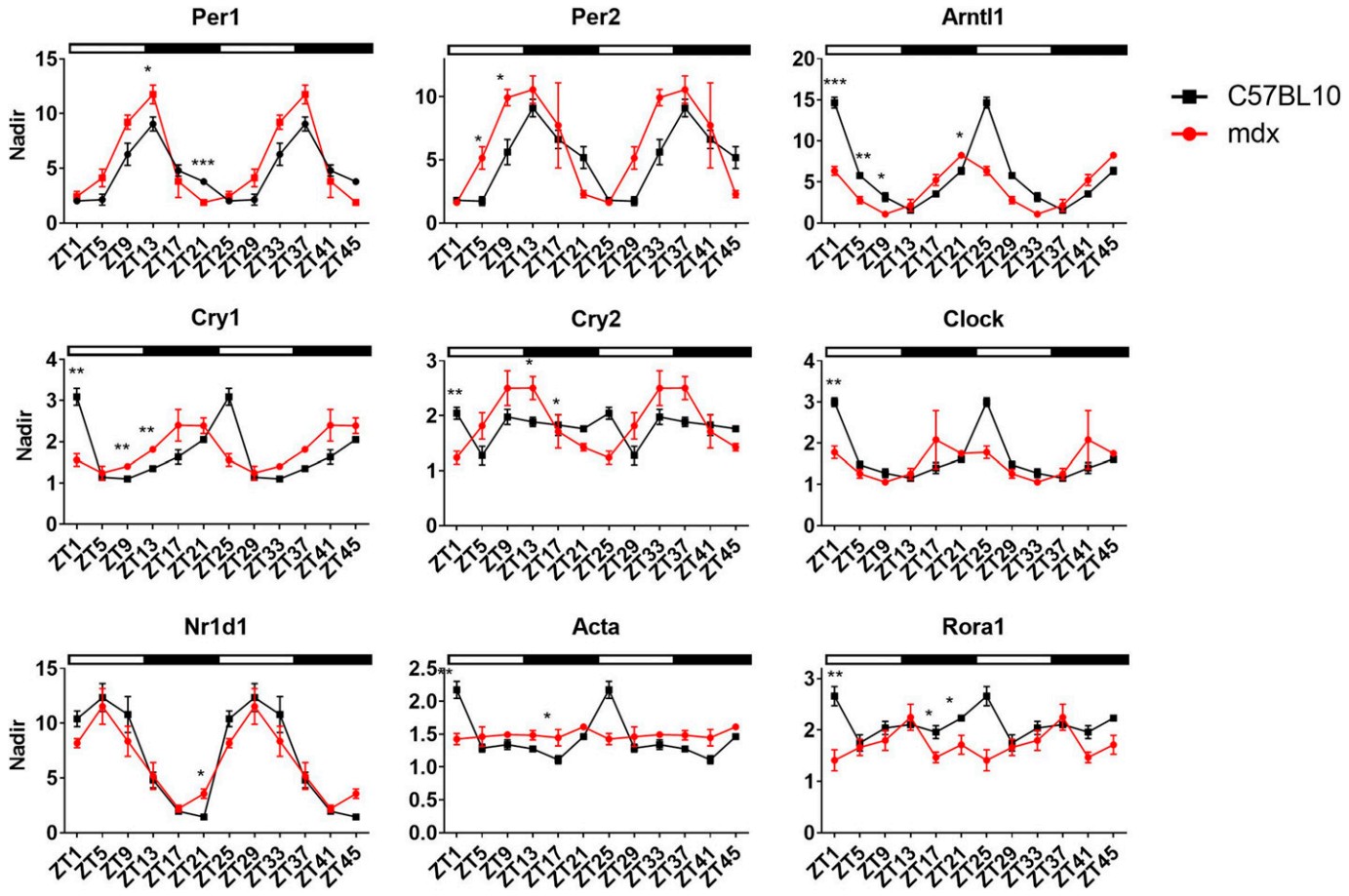

**Figure 4. Altered expression of clock genes in tibialis anterior of mdx mouse model.**
Tissues were collected over a 24-h time course and double plotted (48 h) to better illustrate the oscillation pattern of core clock genes in *tibialis anterior* (TA) of 20 wk old *mdx* animals which were significantly altered compared with C57BL10 (Zeitgeber- ZT). Down-stream targets for RhoA-actin-serum response factor pathway (*Nr1d1*, *Acta*, and *Rora1*) were also altered (*Atp5b* used as housekeeping gene). Light and dark periods represented by outlined (light) and solid bars (dark). The nadir was determined as the minimum value across both genotypes and ZTs and applied to all samples; nadir normalised to one (n = 3–4 for each for each genotype and time-point; two-way ANOVA with Bonferroni post hoc test performed). Mean values reported with SEM; ***P < 0.001, **P < 0.01, *P < 0.05.

These hormones oscillate throughout the day, are highly regulated by sleep–wake cycles, and are governed by SCN signalling (35, 36). This is particularly interesting given that dystrophin is absent in the SCN and may have consequential repercussions on endocrinological processes in DMD patients. It has also been shown that the glucocorticoid, dexamethasone, induces the transcription factor, *Klf15*, with beneficial ergonomic effects on dystrophic muscle (37). Klf15 regulates a multitude of processes including metabolism (38, 39) and nitrogen homeostasis (40), but most importantly does so in a circadian fashion, and has also been shown to be altered in another neuromuscular disorder, spinal muscular atrophy (41). This supports our theory that pharmacological interventions, such as dexamethasone, modulate circadian pathways.

Dystrophic mice were adversely affected by constant light exposure. LL causes arrhythmicity in the long-term, and short-term exposure is a method used for period lengthening which is a common feature of the clock in nocturnal rodents. It is uncertain what effect LL would have in human subjects, but constant dim light has been used (rather than DD) to unmask the central clock,

suggesting light augmentation may be an appropriate therapy for DMD patients. Other therapies including exercise, dietary modification or drugs to mitigate disease pathology (such as lower calcium) (42) may also usefully be investigated in the context of DMD treatment. This study therefore reveals both a hitherto an unanticipated role for dystrophin in peripheral muscle tissues and novel avenues for further research and possible therapeutic intervention applicable to a range of muscular dystrophy disorders.

## Materials and Methods

The datasets generated during this study are available from the corresponding author on reasonable request.

### In vitro *Dmd* knock-down

H2K 2B4 myoblasts were cultured in flasks coated with Matrigel (Corning) and in DMEM culture media supplemented with 2% chick

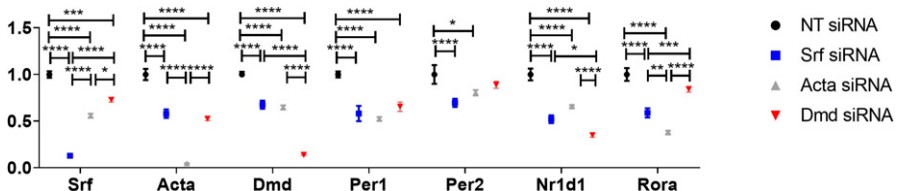

**Figure 5. Abrogation of alternative RhoA-actin-serum response factor (SRF) components leads to a reduction in the expression of SRF target genes.**
H2K 2B4 myotubes were transfected with 100 nM siRNAs targeting the *Srf*, *Acta* and *Dmd* genes (Srf siRNA, blue; Acta siRNA, grey; and Dmd siRNA, red), and a non-targeting (NT siRNA, black) siRNA was used for control. Gene expression data indicate successful knock-down of respective genes (*Srf*, *Acta* and *Dmd*) and down-regulation of pertinent core clock genes (*Per1* and *Per2*) and other down-stream targets for RhoA-actin-SRF (*Nr1d1* and *Acta1*; housekeeping gene—*Gapdh*). Data normalised to NT siRNA; n = 3; two-way ANOVA with Bonferroni post hoc test performed. Mean values reported with SEM; ***$P < 0.001$, **$P < 0.01$, *$P < 0.05$.

embryo extract (Seralabs), 10% fetal calf serum (Gibco), and 1% antibiotics (Gibco); 33°C and 10% $CO_2$. Myoblasts require differentiation into myotubes to produce dystrophin, and therefore differentiation medium containing 5% horse serum (Gibco) was used (DMEM supplemented with 1% antibiotics). Dystrophin was knocked-down using siRNAs targeting the dystrophin transcript (Dharmacon; see Table S1). H2K 2B4 cells were transfected with siRNAs (100 nM) and lipofectamine RNAiMAX (Thermo Fisher Scientific) at day 0 and day 2 of the differentiation process. For RhoA GLISA, F/G actin and MRTF studies, cells were harvested 49 h later, on day 4. Experiments were performed in duplicate, whereby cells were scraped, pooled and further split into three vials for downstream analyses (i.e., protein, F/G-actin). This experiment was repeated three times to attain three biological replicates. Data shown are relative to control siRNA for each experiment. For RT-qPCR study, 49 h after second transfection, media was removed, cells were washed with PBS, and serum free media (DMEM supplemented with 10 mM Hepes and 2% B27- Gibco) was placed in wells. Cells were collected an hour later (CT1), and then every 4 h over a 24-h period. Note: cells were not synchronised with dexamethasone due to the known pharmacological effects observed in dystrophic cells which would likely result in complicated interpretation of the results.

## SRF and actin knock-down

H2K 2B4 myoblasts were again cultured in Matrigel coated flasks using DMEM culture media (2% chick embryo extract, 10% fetal calf serum and 1% antibiotics; 33°C and 10% $CO_2$). Myoblasts were differentiated using medium containing 5% horse serum (Gibco) in DMEM supplemented with 1% antibiotics. Myoblasts were transfected with *Srf* and *Acta* siRNAs (Dharmacon; see Table S1) using Lipofectamine RNAiMAX (Thermo Fisher Scientific) at day 0 and day 2 of the differentiation process. Cells were harvested 49 h after the second transfection, on day 4.

## RNA extraction and RT-qPCR

For patient biopsies, informed consent was obtained from DMD patients' for standard diagnostic purposes, including muscle biopsy. In addition, we obtained informed consent to use muscle biopsies for research activities, according to the Telethon Project N. GGP07011, Ethical approval N. 9/2005, 25 October 2005, by the S. Anna University Hospital Local Ethical Committee, Italy. Patients fasted overnight before the biopsy, which was performed between 8

and 10 am. The control RNA was obtained commercially and consists of two human skeletal muscle samples pooled together (Ambion). RNA was extracted using an RNAeasy kit from QIAGEN as per the manufacturer's instructions.

For mouse muscle and myotubes, total RNA was extracted from tissue or cells using TRIzol (Thermo Fisher Scientific).

All RNA was reverse transcribed using a High Capacity cDNA synthesis kit (Applied Biosystems). The cDNA was run using TaqMan probes sets (Applied Biosystems) and gene specific primer sets for the core clock genes (Integrated DNA Technologies; see Table S1) on the StepOne Real-Time PCR system (Applied Biosystems). Housekeeping genes were determined by running geNorm RT-qPCR normalisation kits with samples from patients (Human geNorm kit-ge-DD-12-hu; Primer Design), cell culture and each mouse model (Mouse geNorm kit-ge-DD-12-hu; Primer Design). *Rpl13a* was the most stable housekeeping gene for Patient biopsies, *Gapdh* for cell culture and dKO samples, and *Atp5b* was the best for *mdx* samples. Tissues were collected over a 24-h time course, and data were double plotted (48 h) to better illustrate the oscillation pattern of core clock genes in muscle. The nadir was determined as the minimum value across all genotypes and ZT's and applied to all samples; nadir normalised to one.

## RhoA activity quantification

RhoA activity was calculated using a RhoA G-LISA Activation Assay kit as per manufacturer's instructions (Cytoskeleton). Briefly, cells were treated with Dmd and NT siRNAs as described above, washed, lysed, and snap-frozen until required. Protein quantification was performed using Precision Red and samples were run on Rho-GTP ELISA.

## F/G actin westerns

Actin protein levels were calculated using a G-actin/F-actin in vivo Assay Biochem Kit as per manufacturer's instructions (Cytoskeleton). Briefly, cells were treated with Dmd and NT siRNAs as described above, were further lysed and underwent ultra-centrifugation to separate fractions. The G-actin supernatant was removed and the F-actin was resuspended in F-actin depolymerisation solution. Samples were run on 10% Bis-Tris gels (Thermo Fisher Scientific) and blotted as per supplier's instruction.

## MRTF westerns

Nuclear and cytoplasmic fractions were collected from tissue culture samples and dKO tissues using the NE-PER kit as described in

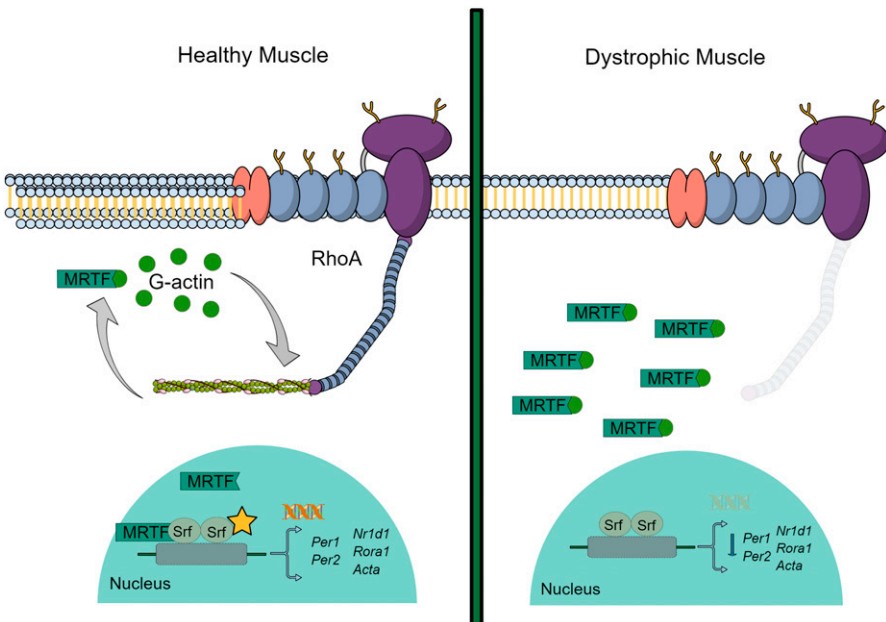

**Figure 6.  Schematic illustrating RhoA-actin-serum response factor (SRF) signalling pathway.**
In healthy muscle, systemic cues activate RhoA in peripheral tissues which in turn regulates diurnal polymerisation (F-actin) and de-polymerisation (globular (G)-actin) of actin. During the de-polymerisation stage, G-actin preferentially binds to myocardin-related transcription factor (MRTF). However, when this shifts to the polymerisation phase, the G-actin pool diminishes and unbound MRTF translocates into the nucleus and influences expression of SRF. SRF activation regulates transcription of SRF target genes including the core clock genes—*Per1* and *Per2*, as well as secondary loop clock genes—*Rora1* and *Nr1d1*, and cytoskeletal genes such as *Acta*. When dystrophin is absent, F-actin is not stable and the cell shifts to the de-polymerised state. G-actin binds to MRTF and therefore SRF transcription is hampered.

manufacturer's instructions (Thermo Fisher Scientific). Samples were run on 10% Bis Tris gels (Thermo Fisher Scientific), blotted onto polyvinylidene difluoride (PVDF) membrane (Millipore) and probed with Mkl1 (ab115319) and Mkl2 (ab191496; Abcam). For nuclear fractions TBP (ab51841; Abcam) was used as loading control, and for cytoplasmic fractions vinculin (hVIN-1; Sigma-Aldrich) and $\beta$-actin were used as loading controls. Blots were visualised using IRDye 800CW goat-anti rabbit IgG or IRDye 800CW goat-anti mouse IgG (LI-COR) on the Odyssey imaging system.

## Dystrophin/utrophin protein quantification

Samples for dystrophin and utrophin westerns were prepared, and protein levels quantified as previously described (43). Briefly, homogenised samples were run on 3–8% Tris-acetate gels (Thermo Fisher Scientific), blotted onto PVDF membrane (Millipore), and probed with DYS1 (Novocastra, dystrophin) or MANCHO antibody (KED laboratory—Oxford, utrophin). Blots were visualised using IRDye 800CW goat-anti mouse IgG (LI-COR) on the Odyssey imaging system. For dystrophin, vinculin (hVIN-1; Sigma-Aldrich) was used as loading control, and for utrophin, the samples were quantified against total protein using Coomassie stain (Sigma-Aldrich).

## Animals housing

All procedures were authorized and approved by the University of Oxford ethics committee and UK Home Office (project licence PDFEDC6F0, protocols 19b1, 2 and 4). Investigators complied with relevant ethics pertaining to these regulations. Procedures were performed in the Biomedical Sciences Unit, University of Oxford. Mice were allowed food and water ad libitum.

## In vivo experiments for gene expression, jetlag, and wheel-running studies

All experiments were performed in male dystrophic C57BL/10ScSn-$Dmd^{mdx}$/J (mdx; JaxLabs), C57B/L10ScSn-Utrn^{tm1Ked} Dmd^{mdx}/J (dKO; Jax-Labs) and control C57BL/10ScSnJ (C57BL10; Envigo and Oxford BSB) mice. For the 24-h ZT study, mice (5 wk old dKO, littermate *mdx* and C57BL10, or 20 wk old *mdx* and C57BL10) were housed under a strict 12:12 h light–dark cycle (LD; 400 lux white light) for 2–3 wk, after which mice were euthanized every 4–6 h over a 24-h time course (n = 3–5 for each genotype and time point). Briefly, mice were culled by cervical dislocation and eyes removed. Animals during the night cycle were culled under dim red light. The TA was dissected, snap frozen and stored at −80° C.

For wheel-running experiments, each mouse was placed in a large cage fitted with a running wheel (n = 6 for each genotype). Activity was recorded using the *Clock*Lab console. Mice were first entrained to a 12:12 h LD cycle at 400lux white light for 2 wk, and then underwent 6-h phase advance entraining, before being placed into constant darkness (DD).

For the jetlag weight study, animals were group housed under a normal 12:12 LD cycle (400 lux white light) for 2 wk to establish stable entrainment (n = 3–4 for each genotype and condition). The LD cycle was then advanced by 6 h every week for 5 wk. Animals were weighed close to ZT13 each week. Animals were culled at the end of the protocol and tissues harvested, weighed and snap frozen.

## Statistics

All statistics were performed using SPSS or GraphPad. For experiments including three or more comparisons, one-way or two-way ANOVA was performed with either Bonferroni or Games–Howell post hoc corrections. For the weight gain experiments, a repeated measures one-way ANOVA was used to investigate the overall

weight gain differences between the groups. For experiments between only two groups, *t* test (two-tailed) was performed.

# Supplementary Information

# Acknowledgements

We would like to thank the Biomedical Sciences facility at Oxford for their care and support of the animals, and Dr Peter Oliver for the use of the controlled light housing units. We would also like to thank Dr Ben Edwards and Prof. Kay Davies for providing utrophin antibody. We thank the Telethon Italy Network of Genetic Biobanks (Dr. Marina Mora) for providing biological samples. Special thanks are due to Duchenne Parent Project Italia Onlus for granting A Ferlini and MS Falzarano DMD diagnostic studies, and Duchenne Parent Project Italy General Grant for DMD cell biobank funding. Work in the laboratory of MJA Wood is supported by the Medical Research Council (MRC programme grant number MRN0248501). CA Betts was supported by the British Heart Foundation and Muscular Dystrophy UK. TLE van Westering was supported by Muscular Dystrophy UK. M Bowerman was an SMA Trust Career Development Fellow when at the University of Oxford. KE Meijboom was funded by the Muscular Dystrophy UK and SMATrust. C Rinaldi is supported by a Career Development Fellowship from the Wellcome Trust (205162/Z/16/Z). Work in the laboratory of MJ Gait was supported by the Medical Research Council (MRC programme number U105178803). JR Counsell is supported by a Wellcome Innovator Award (grant number 210774/Z/18/Z). JE Morgan was supported by Great Ormond Street Hospital Children's Charity (grant number V2137). The support of Muscular Dystrophy UK and the MRC Centre for Neuromuscular Diseases is also gratefully acknowledged. J Meng is supported by the Muscular Dystrophy UK. JR Counsell and JE Morgan are partly funded by the National Institute for Health Research Great Ormond Street Hospital Biomedical Research Centre. The views expressed are those of the authors and not necessarily those of the NHS, the National Institute for Health Research or the Department of Health. Thanks are also due to ERN Euro-NMD (www.ern-euro-nmd.eu) to A Ferlini as member and Chair of the Genetic Task.

## Author Contributions

CA Betts: conceptualization, data curation, formal analysis, funding acquisition, investigation, visualization, methodology, project administration, and writing—original draft, review, and editing.

A Jagannath: conceptualization, data curation, formal analysis, investigation, methodology, and writing—review and editing.

TLE van Westering: data curation, formal analysis, investigation, methodology, and writing—review and editing.

M Bowerman: formal analysis, investigation, and writing—review and editing.

S Banerjee: formal analysis, investigation, and writing—review and editing.

J Meng: investigation and methodology.

MS Falzarano: resources and investigation.

L Cravo: investigation and methodology.

G McClorey: formal analysis and writing—review and editing.

KE Meijboom: formal analysis and investigation.

A Bhomra: formal analysis, investigation, and methodology.

WF Lim: formal analysis, investigation, and methodology.

C Rinaldi: formal analysis.

JR Counsell: investigation and methodology.

K Chwalenia: formal analysis, investigation, and methodology.

E O'Donovan: resources.

AF Saleh: resources.

MJ Gait: resources, supervision, and writing—review and editing.

JE Morgan: formal analysis, supervision, and writing—review and editing.

A Ferlini: resources, supervision, and writing—review and editing.

RG Foster: data curation, formal analysis, and supervision.

MJA Wood: supervision, funding acquisition, and writing—review and editing.

## Conflict of Interest Statement

The authors declare that they have no conflict of interest.

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
