## [Reviewer comments · Life Science Alliance]

Life Science Alliance

Dystrophins involvement in peripheral circadian SRF signalling

Corinne Betts, Aarti Jagannath, Tirsa van Westering, Melissa Bowerman, Subhashis Banerjee, Jinhong Meng, Maria Sofia Falzarano, Lara Cravo, Graham McClorey, Katharina Meijboom, Amarjit Bhomra, Wooi Fang Lim, Carlo Rinaldi, John Counsell, Katarzyna Chwalenia, Elizabeth O'Donovan, Amer Saleh, Michael Gait, Jennifer Morgan, Alessandra Ferlini, Russell Foster, and Matthew Wood
DOI: <https://doi.org/10.26508/lsa.202101014>

Corresponding author(s): Corinne Betts, University of Oxford

Review Timeline:

Submission Date:	2021-01-07
Editorial Decision:	2021-01-08
Revision Received:	2021-03-27
Editorial Decision:	2021-05-18
Revision Received:	2021-07-29
Editorial Decision:	2021-08-02
Revision Received:	2021-08-03
Accepted:	2021-08-04

Transaction Report:

Please note that the manuscript was previously reviewed at another journal and the reports were taken into account in the decision-making process at Life Science Alliance.

Referee #1 Review

Report for Author:

Overall Comments: The manuscript by Betts and colleagues proposes that disruption in dystrophin in muscle (and SCN) leads to altered circadian clock function via the actin/SRF pathway. The intersection of circadian rhythms and muscular dystrophy is just emerging in the scientific community so the topic is novel and timely.

The authors evaluate the regulation of circadian genes in skeletal muscle via dystrophin regulation of F-actin. The authors evaluate SRF circadian target genes in the mdx/utrophin double knockout (dKO) muscles and make inferences that this is due to dystrophin-regulation of F-actin.

The biggest concern with this manuscript is that the authors provide a lot of circumstantial findings with little to no mechanisms linking altered actin/SRF downstream of dystrophin as a clock modifier. This significantly limits any conclusions.

The authors convincingly demonstrate circadian transcript dysregulation and circadian dysregulation in the mdx and dKO mice.

The concepts that are inferred and not demonstrated include: SRF is both A) dysregulated and B) this dysregulation is resulting from dystrophin's regulation of F-actin. Many myogenic factors are dysregulated in DMD, and the authors should be aware of this if they are making the claim that dystrophin's regulation of SRF via F-actin is the causation of circadian gene disruption in muscle.

Major Comments:

1a. Figure 1: questions with regards to their dystrophin siRNA experiments. It was surprising that the group chose to perform all of the figure 1 experiments in primary myotubes knocked down with dystrophin versus myotubes from mdx muscle.

b. There are differences in structural genes/factors between siRNA DMD versus the mdx mice (George Dickson's group highlighted some of these in his works (Seno et al., BMC Genomics, 2010; Seno et al., HMG, 2008). This should be noted.

c. The observation that virtually all core clock genes were downregulated (both positive and negative arm) was a bit surprising and may suggest that the myotubes are not healthy. Can the authors provide 1-2 non clock mRNAs in figure 1g that are not changed?

d. The authors stated that they used a one-tailed student's t-test, this is not justified. Can the authors re-analyze their statistics for Figure 1g and consider ANOVA with posthoc or at least use a two-tailed student's t-test.

e. Human patient DMD muscle lines would be more indicative of SRF-driven dysregulation of circadian target genes. Figure 1h kind of touches on this, but it's confusing to interpret. Unless the authors are trying to suggest dystrophin isoform (e.g. Dp427 versus Dp71, etc.) expression differences in muscle are driving circadian gene differences, I would suggest the authors combine this data to a "normal vs DMD" summary plot with individual data points graphed.

2a. The authors flip between using the mdx mouse and the double knock out (DKO) mice and the rationale for flipping between the two is not clear and the data presentation is confusing. It would be really helpful if they could add Supp Fig 2 to main figure 2, so that readers can visualize the difference in gene expression across control, DKO, and mdx mice.

b. Double-plotting of circadian data (Figure 2a is one example) is inappropriate (see Hughes et al., Guidelines for Genome-Scale Analysis of Biological Rhythms JBR, 2017) . Please redo all the data presented like this and re-run the statistics.

c. the inclusion of the g-actin to f-actin ratio makes no sense other than to document what is already known about dystrophic muscle. These observations provide no mechanistic data about

the SRF pathway and changes in clock genes.

3. Figure 3. The circadian rhythm data are great and the link to dystrophin expression in the SCN is intriguing. However, jetlag experiments and data as presented are of little to no scientific value. The authors missed a significant opportunity with the mice. Did the authors collect muscle samples from these mice? Was there any change in muscle pathology, like centrally nucleated fibers? Did they look at circadian genes or SRF binding abilities to SRF targets? W

4. Overall, there are large scale gaps in the experimental design of this paper. The authors state that it is due to reduced SRF signaling that results in the disruption of circadian genes, but did not directly show this. The appropriate experiment would be to knockdown SRF in myotubes and measure circadian genes to show a direct effect. Moreover, they could also inhibit RhoA to see if it would rescue the SRF expression and thus circadian expression. This is important as previous studies have shown RhoA inhibition in mdx mice ameliorate dystrophic muscle pathologies (Mu et al., 2013 FASEB J; 2017 HMG). The authors also mention a possible role of MRTF downstream of RhoA. Could they have measured bound vs unbound MRTF to demonstrate a significant difference in WT vs mdx muscle that could have supported their explanation?

5. One of the first SRF target genes identified was/is dystrophin (Galvagni et al., Mol. Cell. Biol., 1997). That leads the reviewer to wonder, what is the driving force behind the circadian target dysregulation in mdx muscle. It would suggest that SRF is the driving factor (the authors suggest it is from the F-actin decrease) behind this versus dystrophin loss. siRNA/shRNA knockdown of SRF should result in the decrease in circadian gene expression and the authors should perform this crucial experiment.

Determining the expression levels of SRF in DMD (cells and/or mdx or dKO mice) should be relevant as well.

Minor Comments:

1. Statements like "The dKO mice is considered the optimal dystrophic mouse model" (page 8) ignores some of the more recent DMD mouse models (both CRISPR-generated, and other mdx alleles/strain backgrounds) that have a phenotype somewhat similar to actual dystrophic human pathologies. It is more accurate to state that the dKO is one of the most severely affected DMD mouse models.

2. More descriptions with regards to the SRF muscle KO mice (their phenotypes) and muscle roles (e.g. Djemai et al., Eur. J. Trans. Myol., 2018) demonstrating early muscle weakness and sarcopenia. I would recommend the authors consider what stretch force dynamic regulation of SRF/dystrophin might be driving alterations of circadian mRNAs.

3. The Gao et al. Exp. Cell. Res., 2020 manuscript just came out demonstrating that mdx mice have a defect in Bmal1 targets. This paper probably should be referenced.

Referee #2 Review

Report for Author:

In this study, C.A. Betts investigates the impact of dystrophin loss on circadian locomotion behavior in mice and on circadian signaling in a peripheral tissue, skeletal muscle. By using cultured muscle cells and mice lacking dystrophin or both dystrophin/utrophin, the authors suggest that dystrophin loss causes peripheral circadian deficits through alterations of RhoA-actin-Srf signaling.

This study is based on the observations showing that mdx mice (lacking dystrophin) display an overall perturbed rest-activity behavior and body weight upon the modifications of light cycles (DarkDark, LightLight, jet lag), and on data showing an alteration of F/G actin ratio and possibly of Srf activity in cultured myotubes and in muscle tissue. The link between these two observations is only correlative. In addition several experiments are missing to further decipher whether the phenotype observed in the dystrophic mice is due to alterations of the central clock (SCN) and/or to alterations in the muscle tissue. The peripheral alteration of actin-Srf signaling is as well poorly demonstrated.

In order to straighten the following statement "Dystrophin regulates peripheral circadian SRF signalling" (which is the title of the article), the authors may need to perform additional experiments because as such the data presented do not support their conclusion.

Specific major concerns:

1. In Figure 1 are presented data on myotubes in which Dmd gene expression is down regulated. It seems that the total level of actin is decreased in Dys siRNA samples, is it true after quantification? If F/G ratio decreased and Srf activity decreases, then MrtfA/B should be less nuclear. Is it the case? IF experiments or WB on cytoplasmic/nuclear extracts using antibodies against MrtfA/B should be performed to get additional insights on the decrease of this signaling pathway.
2. F/G actin ratio has been quantified in muscle tissue only at a time point (ZT not indicated, Figure 2). It is very important to investigate whether F/G actin ratio and nuclear Mrtf accumulation exhibit diurnal changes in muscle tissue and not show only one time point. Is this oscillation altered in dystrophic mice?
3. Part of the effect observed on general behavior of mdx mice could be attributed to systemic cues from the central clock. In order to demonstrate whether peripheral disorders of mdx mice may contribute as well, restoration of Dystrophin expression only in muscle tissue could be performed using for instance AAV-miniDystrophin or activation of actin-Mrtf-Srf pathway could be achieved using constitutively active Mrtf or Srf derivatives (AAV viral vectors).
4. Is there a peripheral oscillation independently of central clock? A recent paper from K. Esser lab, showed that after synchronization, C2C12 myotubes exhibit an endogenous rhythm of core clock genes such as Per1, Per2, Bmal1, Cry1 (Kemler et al, 2020). It would be nice to assess after synchronization whether there is a perturbation of this clock in cultured myotubes expressing or not dystrophin. Then the inhibition of actin-Mrtf-Srf pathway (siRNA or CCG drug) should affect this rhythm in control myotubes and activation of this pathway should restore this clock in myotubes lacking dystrophin.
5. Mrtf should appear in the nucleus in the healthy muscle (Figure 4 schematic illustration)
6. Why is there less overall actin and a more profound effect on F/G ratio in muscle tissue from mdx (FigS2) as compared to double dystrophin/utrophin mutants (Fig2)?

Specific minor comments:

The Introduction is too short and a lot of informations are missing on the clock in skeletal muscle, the role of actin/Mrtf/srf pathway in skeletal muscle etc...

Reference

Kemler D, Wolff CA, Esser KA. Time-of-day dependent effects of contractile activity on the phase of the skeletal muscle clock. *J Physiol.* 2020 Sep;598(17):3631-3644.

Referee #3 Review

Report for Author:

In this manuscript, the authors describe circadian clock defects in the absence of the protein Dystrophin. They first use a myoblast cell line (H2K 2B4) in which they disrupt dystrophin gene expression, and conclude that the expression of circadian genes, and the DNA binding of the circadian clock transcription factor SRF, is disrupted in cells lacking dystrophin. They analyze patient samples for the expression of circadian clock genes, and they monitor circadian behavior in the mdx mouse model. Although the conclusions of their studies are undoubtedly novel, there are many gaps in the experimental design and in the data analyses. The manuscript, in its present form, falls short of meeting the high standards of EMBO Journal.

Major points:

1. The authors use a cell line (H2K 2B4) and transfect them to knock-down Dmd expression. The authors already have 2 good mice models that are dystrophin-deficient (mdx and mdx-Utrn). The authors should use myoblasts derived from these mice to perform the experiments in Fig 1 a-g, as they would be an appropriate muscular dystrophy model.
1. The authors describe a signalling pathway in which dystrophin perturbs the F-G actin ratio. In Figures 1c and 2b, the authors present cropped Western blots to support their conclusions, which is not acceptable for journal publication. Please include uncropped Western blots with all the samples loaded side by side on the gel, and include a loading control blot. The quantification suggests a 50% decrease. If such a drastic decrease is observed, this should be also visible by a simple phalloidin staining.
2. There are many concerns about the SRF ChIP data in Figure 1f. First, the authors used a "beads-only" control, which is not adequate. Robust ChIP experiments include an IgG control. The data should be presented as %input and the IgG control lane should be visible on the graphs. The addition of negative DNA loci regions (not bound by SRF) should be included as well. There is also a typo in the Methods section, where the authors state that they used 5mg of anti-SRF antibody (5ug?). Finally, performing the SRF ChIP in DMD patients and in primary myoblasts derived from mdx mice would definitely reinforce the authors conclusions.
3. In Figures 1h and S3c, the authors analyze gene expression in DMD patients. They use only one (1) healthy control sample, is insufficient as human populations are extremely variable. Typically an n of 10 is required for humans. The authors should also report which reference (housekeeping) genes were used for the analyses, in the Figure legends as well as in the Methods section.
4. In Figure 1, circadian clock genes (Cry1, Cry2, Arntl, Clock, Acta and Rora1) are down-regulated in the Dmd-siRNA treated cells, as well as in patients. However, in Figure 2, the same genes are up-regulated in mdx mice, which is very puzzling and questions the use of the mdx mouse model in further testing circadian rhythm (Fig 3). The authors do not comment, discuss or explain this discrepancy.
5. In Figure 3, there is a lack of concordance in the data presentation between the LD, DD and LL conditions. For example, in LD condition, the authors present the number of bouts and phase angle, while in LL they plot the activity and free running. The authors should present the data in a more

homogenous manner, i.e. by plotting the number of bouts, phase angle, free running period and activity, in all the conditions tested.

6. In figure 3i, the authors compare mice that are weight-matched, whereas in Figure 3j the mice are aged-matched. Again, the authors should present the same sets of controls for all the experiments.

7. In the Discussion section, the authors state that melatonin treatment improves the condition of DMD patients. Is it the case in mdx mice? This experiment is easy to perform and would definitely strengthen the manuscript.

January 8, 2021

Re: Life Science Alliance manuscript #LSA-2021-01014-T

Dr. Corinne Anne Betts
University of Oxford
Paediatrics
Le Gros Clarke Building
South Parks Road
Oxford OX1 3QX
United Kingdom

Dear Dr. Betts,

Thank you for submitting your manuscript entitled "Dystrophin regulates peripheral circadian SRF signalling" to Life Science Alliance.

For a brief overview, the manuscript was previously reviewed at one of our alliance journals. The editors of the previous journal shared the manuscript and accompanying reviewers' comments with Life Science Alliance (LSA), with the authors' permission. LSA editors found the topic to be interesting and timely and offered the authors an opportunity to pursue their publication further at LSA, provided the following requests are addressed in the revision,

EXPERIMENTAL REQUESTS:

- + addressing Rev 1 pts 1c, 4, and 5 are critical to the main message put forth by this manuscript and these experiments should be included in the revised manuscript.
- + addressing Rev 2 pts 1 and 2 and Rev 3 pt 3 are critical to make the data robust and should be included in the revised manuscript. We would understand if getting more human control samples (to address Rev 3, pt 3) might be difficult, please feel free to discuss this point with Shachi directly, if needed.
- + In response to Rev 3 pt 2, about the concerns with SRF ChIP data - we suggest you present the data as %input and include the negative DNA loci regions (not bound by SRF) .
- + address all the minor concerns raised by the three reviewers.

ADDRESS VIA TEXT CHANGES AND ADDITIONAL STATISTICAL ANALYSIS:

- + Rev 1 pts 1d, 1e, 2a - c, Rev 2 pts 5 and 6, and Rev 3 pt 1 (the second on the list) and pts 4 - 6 can be addressed by making the necessary textual changes, adding further discussion and better statistical analysis in the revised manuscript.

POINTS THAT DO NOT NEED TO BE ADDRESSED EXPERIMENTALLY, BUT SHOULD BE DISCUSSED:

- + All 3 reviewers have requested that some of the data from Figure 1 be re-done with myotubes from mdx mice. These experiments will NOT be required for LSA.
- + performing additional experiments with mice subjected to jetlag (Rev 1 pt 3), restoring Dystrophin expression only in the muscle tissue with AAV to see if it restores the some or all phenotypes (Rev 2 pt 3), assessing whether there is perturbation of the endogenous rhythm of core clock genes in C2C12 myotubes (Rev 2 pt 4), and performing melatonin treatment of mdx mice (Rev 3 pt 7) would

NOT be required for LSA. However, we do encourage you to discuss these points in the manuscript.
+ We agree with Rev 3 that an IgG control is ideal for ChIP experiments (Rev 3 pt 2), and thus we encourage you to clearly indicate in the manuscript that this control was not included in the experiment. Performing the ChIP experiment in primary myoblasts from mdx mice or DMD patients would NOT be required for further consideration at LSA.

We encourage you to submit a revised version of this manuscript addressing the points mentioned above, and including a point-by-point response to all the comments from the reviewers at the previous journal.

Thank you for this interesting contribution to Life Science Alliance. We are looking forward to receiving your revised manuscript.

Sincerely,

Shachi Bhatt, Ph.D.

Executive Editor

Life Science Alliance

<https://www.lsajournal.org/>

-- High-resolution figure, supplementary figure and video files uploaded as individual files: See our

detailed guidelines for preparing your production-ready images, <https://www.life-science-alliance.org/authors>

B. MANUSCRIPT ORGANIZATION AND FORMATTING:

We would like to thank the reviewers for their time, positive feedback and excellent suggestions to improve our manuscript entitled "*Dystrophin regulates peripheral circadian SRF signalling*". We believe we have taken on-board all of the comments and addressed the questions raised by the three expert Reviewers and appreciate that this has significantly improved the quality of our manuscript and strength of its message for which we are very grateful.

Referee #1:

1c. The observation that virtually all core clock genes were downregulated (both positive and negative arm) was a bit surprising and may suggest that the myotubes are not healthy. Can the authors provide 1-2 non clock mRNAs in figure 1g that are not changed?

We thank the reviewer for this suggestion. We have repeated the experiment and looked at cells over a 24hr time-course to assess changes in gene expression. These results can be found in new Figure 1f and correlate closely with the dKO gene data (Fig 2a).

This dataset is described on page 6: 'To illustrate circadian oscillation patterns of core clock genes involved in the transcriptional auto-regulatory feedback loop following abrogation of dystrophin, cells were collected every 4 hrs over a 24 hr period, and indicate significant alterations (Fig 1F). *Per1*, *Per2* and *Clock* expression were significantly down-regulated at certain time points, whilst *Cry2* and *Arntl1* were upregulated. Expression of other downstream SRF target genes in the actin-cascade, *Nr1d1* and *Acta*, were also significantly lower in dystrophin deficient samples'.

d. The authors stated that they used a one-tailed student's t-test, this is not justified. Can the authors re-analyze their statistics for Figure 1g and consider ANOVA with posthoc or at least use a two-tailed student's t-test.

We agree with the reviewer that a two-tailed student's t-test was more appropriate. However, as Figure 1g has now been replaced with new Figure 1f, the re-analysis is no longer applicable.

e. Human patient DMD muscle lines would be more indicative of SRF-driven dysregulation of circadian target genes. Figure 1h kind of touches on this, but it's confusing to interpret. Unless the authors are trying to suggest dystrophin isoform (e.g. Dp427 versus Dp71, etc.) expression differences in muscle are driving circadian gene differences, I would suggest the authors combine this data to a "normal vs DMD" summary plot with individual data points graphed.

We thank the reviewer for the comment. However, as full-length dystrophin (Dp427) is the isoform that tethers F-actin, we believe that representing each mutation/deletion individually is more informative and is of greater interest to the DMD Scientific community.

2a. The authors flip between using the mdx mouse and the double knock out (DKO) mice and the rationale for flipping between the two is not clear and the data presentation is confusing. It would be really helpful if they could add Supp Fig 2 to main figure 2, so that readers can visualize the difference in gene expression across control, DKO, and mdx mice.

We apologise if the use of two models was confusing and have made the following changes and clarifications:

1. We have elaborated on our rationale for showing the *mdx* gene expression data on page 8, paragraph 2: 'Whilst the dKO mouse model closely recapitulates the dystrophic phenotype in patients and is a good molecular model for DMD, it's severe phenotype including reduced lifespan (approx. 5-8 weeks) and marked reduction in activity, preclude extensive locomotor behaviour studies. As such, the less affected dystrophic model, *mdx*, was utilised for the extensive battery of locomotor tests'.
2. We have moved the *mdx* gene expression data to the main manuscript and is now presented in **new Figure 4**.
3. We discuss the data from the 2 mouse models in greater detail and postulate as to the rationale for differences observed between them on page 10, paragraph 3: 'Whilst the *mdx* gene dataset differs to the dKO and H2K 2B4 myotube models, we regard the dKO model with greater esteem given its phenotype and correlation with patient disease progression, which is supported by the biopsy data (Fig 1H). It further correlates closely with cell-culture data in which dystrophin is specifically abrogated (Fig 1F). However, in order to illustrate locomotive aberrations, and systemic cues with the SCN, it was imperative we look in the milder *mdx* model. The remarkable lifespan and generally mild phenotype of *mdx* mice is poorly understood, but it is likely due to multiple compensatory events that ensue and may account for the variances observed between the dystrophic models. As multiple inputs can regulate the clock, it is difficult to predict how the clock will react when one input is removed. *In vivo* complexities in the form of protein or signalling interactions with serum, hormones (particularly glucocorticoids as *Per2* has a glucocorticoid receptor-binding site) and neurological signals or mechanisms triggered to compensate for disturbed RhoA-actin-SRF pathway may be involved. One such compensatory mechanism is upregulation of utrophin, a homologue of dystrophin, which is knocked-out in dKO but present in *mdx* mice. Utrophin has been shown to bind and maintain F-actin polymerisation and therefore seems an obvious candidate in the RhoA-actin-SRF signalling cascade.'

b. Double-plotting of circadian data (Figure 2a is one example) is inappropriate (see Hughes et al., Guidelines for Genome-Scale Analysis of Biological Rhythms JBR, 2017). Please redo all the data presented like this and re-run the statistics.

We thank the reviewer for their comment. We have double-plotted the data to better illustrate the oscillation pattern, as previously published in PNAS and Cell: <https://www.pnas.org/content/116/37>, <https://www.sciencedirect.com/science/article/pii/S0092867418307992#fig2>. Furthermore, the suggested experiments entails unreasonable costs, time and ethical implications, as we would require at least 24-36 animals per study.

c. the inclusion of the g-actin to f-actin ratio makes no sense other than to document what is already known about dystrophic muscle. These observations provide no mechanistic data about the SRF pathway and changes in clock genes.

We agree with the reviewer that the altered F/G-actin ratio has previously been shown in dystrophic animals. Here, we add to this knowledge by showing for the first time changes in dystrophin-depleted cell models (Fig 1D) as well as diurnal fluctuations in the F and G actin levels (Fig 2B; in control animals).

4. Overall, there are large scale gaps in the experimental design of this paper. The authors state that it is due to reduced SRF signaling that results in the disruption of circadian genes, but did not directly show this. The appropriate experiment would be to knockdown SRF in myotubes and measure circadian genes to show a direct effect.

We thank the reviewer for this suggestion and have now included a new **Figure 5** showing *Srf* and another cascade component, *Acta*, knock-down. SRF knock-down in myotubes results in very similar gene expression patterns compared to *Dmd* knock-down. This is described on page 12 paragraph 1: 'To assess the RhoA-actin SRF cascade further, and determine whether abrogation of other upstream components of the pathway, such as actin, or indeed SRF itself, results in similar changes in the expression of target genes, siRNAs were used to specifically knock-down *Srf* and *Acta* in H2K 2B4 myoblasts. In order to compare with dystrophin knock-down experiments, differentiated myotubes were transfected twice with 100nM *Srf* and *Acta* siRNAs, alongside *Dmd* transfected myotubes, and collected 49 hours following the second transfection, thereby representing CT1. Myotubes were also treated with lower concentrations of siRNA to confirm gene expression was stable and that myotubes were healthy (Fig S4). Interestingly, *Srf*, *Acta* and *Dmd* down-regulation, appear to reciprocally modulate each other resulting in lower expression of all genes for all cohorts ie. *Srf* knock-down results in lower *Dmd* and *Acta* gene expression and *vice versa* (Fig 5). Additionally, all siRNA treatment groups resulted in reductions of RhoA-actin-SRF target genes, *Per1* and *Per2*, and *Nr1d1* and *Rora*. Together, this illustrates how intertwined and mutually dependent these genes are in maintaining homeostatic balance of the RhoA-actin-SRF cascade.'

Moreover, they could also inhibit RhoA to see if it would rescue the SRF expression and thus circadian expression. This is important as previous studies have shown RhoA inhibition in mdx mice ameliorate dystrophic muscle pathologies (J; 2017 HMG).

We apologise that our original graphical representation of the cascade (now Fig 6) was unclear, leading the Reviewer to believe that the G-actin subunits (in green) represented RhoA, and therefore by downregulating RhoA would allow SRF activation. However, this is contrary to our hypothesis which postulates that RhoA activity facilitates polymerisation of actin into the filamentous form (F-actin) and thereby inhibiting this activity would lead to more G-actin which binds to MRTF. In MRTFs bound state it is unable to enter the nucleus and regulate SRF expression (Liu, PMID: 12600823) and therefore we would observe a similar cascade to that observed in dystrophic muscle. Mu *et al* indeed observed therapeutic benefits of RhoA inhibition in dystrophic animals, which they determine to be due to reduced expression of BMPs and inflammation, and not related to the SRF activation. We have however looked at another upstream component of the RhoA-actin-SRF cascade, and performed *Acta* knock-down studies in parallel to *Dmd* and *Srf* knock-down studies (new Fig 5). Again, we observe similar gene expression patterns when these alternative components are abrogated in this cascade. Importantly, we have provided a **new and updated Fig 6** to better represent the RhoA and G-actin components.

Old Fig 4

New Fig 6

The authors also mention a possible role of MRTF downstream of RhoA. Could they have measured bound vs unbound MRTF to demonstrate a significant difference in WT vs mdx muscle that could have supported their explanation?

We thank the reviewer for this suggestion and have provided **new Figures** to show MRTF levels in myotubes (Fig 1E) and diurnal fluctuations of the protein in dKO TA muscle (Fig 2C).

We have also updated the text:

Page 6: 'Additionally, there appears to be greater cytoplasmic MRTF accumulation under dystrophic conditions (not significantly different; Fig 1E).'

Page 7, paragraph 2: 'In dKO animals, the levels of nuclear MRTF protein trends lower (Fig 2C, ZT1), and MRTF cytoplasmic protein levels were significantly lower (ZT13) compared to control animals, which together suggests there is less total MRTF in dKO animals.'

5. One of the first SRF target genes identified was/is dystrophin (Galvagni et al., Mol. Cell.

Biol., 1997). That leads the reviewer to wonder, what is the driving force behind the circadian target dysregulation in mdx muscle. It would suggest that SRF is the driving factor (the authors suggest it is from the F-actin decrease) behind this versus dystrophin loss. siRNA/shRNA knockdown of SRF should result in the decrease in circadian gene expression and the authors should perform this crucial experiment.

We thank the reviewer for their comment and believe that the upstream component of the MRTF-actin-SRF cascade, dystrophin, is imperative to its function and that absence of dystrophin disrupts signalling with knock-on effects on SRF activity. We agree with the reviewer that SRF is integral for dystrophin function also, which illustrates their cyclical and integral relationship. We have elaborated on the introduction to include more information on SRF and dystrophin interactions in the manuscript; page 4 paragraph 2 'Indeed SRF is a pivotal nuclear transcription factor, regulating over 200 target genes that are predominantly involved in cell-growth, migration, cytoskeletal organisation and myogenesis, and one of the earliest SRF target genes to be identified was *Dmd*. This integral relationship, combined with the understanding that they operate via a feed-back loop, intimates that the absence of dystrophin would have serious implications on SRF regulation. Interestingly, studies designed to mimic age related sarcopenia by disrupting skeletal muscle SRF expression resulted in atrophy, fibrosis, lipid accumulation and disturbed regeneration which are all hallmarks of the DMD phenotype further supporting the cyclical nature and mutual dependence.'

Determining the expression levels of SRF in DMD (cells and/or mdx or dKO mice) should be relevant as well.

We thank the reviewer for this suggestion and have now included *Srf* gene expression levels in myotubes (Fig 1F) and dKO mice (Fig 2A).

Minor Comments:

1. Statements like "The dKO mice is considered the optimal dystrophic mouse model" (page 8) ignores some of the more recent DMD mouse models (both CRISPR-generated, and other mdx alleles/strain backgrounds) that have a phenotype somewhat similar to actual dystrophic human pathologies. It is more accurate to state that the dKO is one of the most severely affected DMD mouse models.

We agree with the Reviewer and have removed the use of the word 'optimal' from the text; page 8 paragraph 2: 'The dystrophin-utrophin knockout (dKO) model presents with a severe phenotype that closely recapitulates disease in patients, specifically severe progressive muscular dystrophy, premature death and a plenitude of physiological and molecular aberrations'.

More descriptions with regards to the SRF muscle KO mice (their phenotypes) and muscle roles (e.g. Djemai et al., Eur. J. Trans. Myol., 2018) demonstrating early muscle weakness and sarcopenia. I would recommend the authors consider what stretch force dynamic regulation of SRF/dystrophin might be driving alterations of circadian mRNAs.

We agree with the reviewer that reference to SRF-KO models should be included in the manuscript and have now included this; page 4 paragraph 2: 'Interestingly, studies designed to mimic age related sarcopenia by disrupting skeletal muscle SRF expression resulted in atrophy, fibrosis, lipid accumulation and disturbed regeneration which are all hallmarks of the DMD phenotype further supporting the cyclical nature and mutual dependence.'

The Gao et al. Exp. Cell. Res., 2020 manuscript just came out demonstrating that mdx mice have a defect in Bmal1 targets. This paper probably should be referenced.

We thanks the reviewer for the suggestion and have now included the reference, of which we were not aware of the manuscript at the time of submission; page 4 paragraph 1: 'These rhythmic genes are involved in many central processes such as myogenesis, muscle lipid utilisation, protein metabolism and organisation of myofilaments, and very recently a key circadian gene, *Bmal1* (*Arntl*), was shown to be involved in impaired myogenicity in muscle of dystrophic mice.'

Referee #2:

1. In Figure 1 are presented data on myotubes in which Dmd gene expression is down regulated. It seems that the total level of actin is decreased in Dys siRNA samples, is it true after quantification? If F/G ratio decreased and Srf activity decreases, then MrtfA/B should be less nuclear. Is it the case? IF experiments or WB on cytoplasmic/nuclear extracts using antibodies against MrtfA/B should be performed to get additional insights on the decrease of this signaling pathway.

We thank the reviewer for their comment. However, considering the experimental approach used (commercially available kits, standardisation, independent repetition of the experiments and reproducibility of the results), we are confident of the results demonstrating that F/G actin ratios are significantly lower under dystrophic conditions compared to healthy/control conditions.

We agree with the reviewer regarding the value of showing the inclusion of nuclear/cytoplasmic MRTF levels and have now included this for myotubes (new Fig 1E) and mouse models (new Fig 2C).

We have also updated the text;

Page 6: 'Additionally, there appears to be greater cytoplasmic MRTF accumulation under dystrophic conditions (not significantly different; Fig 1E).'

Page 7, paragraph 2: 'In dKO animals, the levels of nuclear MRTF protein trends lower (Fig 2C, ZT1), and MRTF cytoplasmic protein levels were significantly lower (ZT13) compared to control animals, which together suggests there is less total MRTF in dKO animals.'

2. F/G actin ratio has been quantified in muscle tissue only at a time point (ZT not indicated, Figure 2). It is very important to investigate whether F/G actin ratio and nuclear Mrtf accumulation exhibit diurnal changes in muscle tissue and not show only one time point. Is this oscillation altered in dystrophic mice?

We thank the reviewer for this suggestion and have now included ZT1 and ZT13 time points (diurnal fluctuations) for nuclear/cytoplasmic MRTF and F/G actin levels for the mouse models in new Fig 2B and Fig 2C. We have also updated the text, page 7, paragraph 2: 'It is important to note that F/G actin ratios exhibited diurnal changes in skeletal muscle of healthy mice, and MRTF cytoplasmic fraction levels appear to oscillate also.'

5. Mrtf should appear in the nucleus in the healthy muscle (Figure 4 schematic illustration)

We apologise for the confusion. We were trying to indicate that MRTF **moved** into the nucleus and agree that this was unclear so have updated the figure to illustrate this better.

Old Fig 4

New Fig 6

6. Why is there less overall actin and a more profound effect on F/G ratio in muscle tissue form mdx (FigS2) as compared to double dystrophin/utrophin mutants (Fig2)?

We thank the reviewer for their question. A number of factors could be at play here such as the age of animals, running and probing samples on different membranes, slight variances in extraction (please note all samples for each experiment were extracted and run at the same time). We can however show data which includes age matched, littermate *mdx* and dKO animals (extracted at the same time as dKO/C57BL10 samples) which was not included in the original manuscript. This shows that dKO animals have lower F/G-actin ratios then both *mdx* and C57BL10 animals. We have now included this graph (Fig S2) and updated the text with the following; page 7 paragraph 2: 'F/G-actin ratios were significantly lower in 5 week old dKO animals compared to control animals (Fig 2B) and littermate *mdx* animals (Fig S2), indicating a more profound impact on F-actin in dKO compared to the less effected *mdx* model.'

Specific minor comments:

The Introduction is too short and a lot of informations are missing on the clock in skeletal muscle, the role of actin/Mrtf/srf pathway in in skeletal muscle etc...

We agree with the reviewer and have now rewritten the introduction. Please see pages 4&5 for the substantially revised full text.

Referee #3:

1. The authors describe a signalling pathway in which dystrophin perturbs the F-G actin ratio. In Figures 1c and 2b, the authors present cropped Western blots to support their conclusions, which is not acceptable for journal publication. Please include uncropped Western blots with all the samples loaded side by side on the gel, and include a loading control blot.

We thank the reviewer for the suggestion and have now included all uncropped westerns blots in the 'figure source data' section, as set out in the journal guidelines: Western Fig 1, Western Fig 2, Western Fig 3, Western Fig S3.

2. There are many concerns about the SRF ChIP data in Figure 1f. First, the authors used a "beads-only" control, which is not adequate. Robust ChIP experiments include an IgG control. The data should be presented as %input and the IgG control lane should be visible on the graphs. The addition of negative DNA loci regions (not bound by SRF) should be included as well. There is also a typo in the Methods section, where the authors state that they used 5mg of anti-SRF antibody (5ug?).

We thank the reviewer for their suggestions and have now presented the data as % input and included the negative antibody controls for each sample in Figure 1G.

We also thank the Reviewer for bringing the typo to our attention and have now rectified this.

Updated text on page 6: 'SRF chromatin-immunoprecipitation (ChIP) combined with qRT-PCR revealed reduction of occupancy of SRF on the *Nr1d1* regulatory region and partial reduction on *Per2*, which are two established targets of the MRTF-SRF cascade, thereby suggesting impaired SRF signalling (Fig 1G, see Fig S1 for SRF-binding motifs).'

Updated methods page 16: 'ChIP was performed using Dynabeads Protein G (ThermoFisher, 1003D) and 5µg of anti-SRF antibody (ab53147; Abcam) as previously described. The immunoprecipitated (SRF antibody in immunoprecipitation (IP) reaction), control (beads only) and input DNAs were used as templates for qPCR, with SYBR Green PCR Master Mix (Applied Biosystems) and gene-specific primers (Table S1). The % Input was calculated for each gene region by subtracting the input from the IP sample (delta Ct) and then performing a power calculation ($100 \times 2^{\Delta Ct}$). The negative background signal (as determined by the 'no antibody' control) was calculated in a similar manner.'

3. In Figures 1h and S3c, the authors analyze gene expression in DMD patients. They use only one (1) healthy control sample, is insufficient as human populations are extremely variable. Typically an n of 10 is required for humans. The authors should also report which reference (housekeeping) genes were used for the analyses, in the Figure legends as well as in the Methods section.

We thank the reviewer for their comment. The 'healthy control sample' was attained by pooling muscle biopsies from 2 healthy volunteers, valuable samples that are extremely difficult to attain, thus limiting our ability to acquire more. We acknowledge that it was not clearly stated in the text that the healthy sample was acquired from 2 volunteers and have now included this in the methods section (page 17) and figure legend (page 25). We have also included the housekeeping gene information in the Figure legends and Methods section for all RT-qPCR data.

4. In Figure 1, circadian clock genes (Cry1, Cry2, Arntl, Clock, Acta and Rora1) are down-regulated in the Dmd-siRNA treated cells, as well as in patients. However, in Figure 2, the same genes are up-regulated in mdx mice, which is very puzzling and questions the use of the mdx mouse model in further testing circadian rhythm (Fig 3). The authors do not comment, discuss or explain this discrepancy.

We thank the reviewer for their comment and have now repeated the experiment in Figure 1, and looked at the oscillation patterns of core clock genes in dystrophic cells over a 24hr time course, observing very similar gene expression patterns to dKO mice. We consider the dystrophic myotubes and dKO mice to be the most reliable models given the specific down-regulation of dystrophin in the myotubes and the similar phenotype observed in dKO animals and DMD patients. However, the severity of the dKO phenotype, which leads to premature death and low activity, prohibits the extensive battery of locomotor tests required to assess endogenous clock alterations and SCN signalling. As such, the less affected model- *mdx*- was utilised to provide a comprehensive and transparent overview of all models used. Whilst the *mdx* and dKO/dystrophic cell gene datasets do differ, we discuss the rationale for this on page 10, paragraph 3: 'Whilst the *mdx* gene dataset differs to the dKO and H2K 2B4 myotube models, we regard the dKO model with greater esteem given its phenotype and correlation with patient disease progression, which is supported by the biopsy data (Fig 1H). It further correlates closely with cell-culture data in which dystrophin is specifically abrogated (Fig 1F). However, in order to illustrate locomotive aberrations, and systemic cues with the SCN, it was imperative we look in the milder *mdx* model. The remarkable lifespan and generally mild phenotype of *mdx* mice is poorly understood, but it is likely due to multiple compensatory events that ensue and may account for the variances observed between the dystrophic models. As multiple inputs can regulate the clock, it is difficult to predict how the clock will react when one input is removed. *In vivo* complexities in the form of protein or signalling interactions with serum, hormones (particularly glucocorticoids as *Per2* has a glucocorticoid receptor-binding site) and neurological signals or mechanisms triggered to compensate for disturbed RhoA-actin-SRF pathway may be involved. One such compensatory mechanism is upregulation of utrophin, a homologue of dystrophin, which is knocked-out in dKO but present in *mdx* mice. Utrophin has been shown to bind and maintain F-actin polymerisation and therefore seems an obvious candidate in the RhoA-actin-SRF signalling cascade.'

5. In Figure 3, there is a lack of concordance in the data presentation between the LD, DD and LL conditions. For example, in LD condition, the authors present the number of bouts and phase angle, while in LL they plot the activity and free running. The authors should

present the data in a more homogenous manner, i.e. by plotting the number of bouts, phase angle, free running period and activity, in all the conditions tested.

We thank the reviewer for their suggestion. However, we are unable to look at 'free running' in LD as this is a measurement of internal clock, which can only be determined in DD and LL (as light is Zeitgeber during LD), and similarly we cannot measure 'phase angle' data when in LL (as this measures when mice become active in the dark phase). We have however included all other data collected including 'activity in DD', activity following light pulses and jet-lag. The updated text states on page 8 paragraph 2 'Interestingly, during the light phase of LD, activity of *mdx* mice was markedly reduced, and they exhibited delayed onset into dark phase (phase angle). Following 6-hour phase advance bouts, *mdx* mice were capable of re-entraining to the shifted cycle in a similar manner to control animals (Fig 3D). Animals were then placed in DD, where *mdx* animals again indicated a delayed onset (phase angle) on release into dark (Fig 3E), suggesting that their endogenous clock may be out of phase. Again endurance of *mdx* animals was maintained during DD and they ran for a similar period compared to C57BL10 animals, however their free running period was significantly shorter (Fig 3F; *mdx* run half an hour shorter). During the DD phase, mice received a light pulse 4-hours after they started exercise (CT16), and *mdx* mice displayed no difference in the ability to shift the clocks phase in response to this nocturnal light compared to C57BL10 animals (Fig 3G).'

6. In figure 3i, the authors compare mice that are weight-matched, whereas in Figure 3j the mice are aged-matched. Again, the authors should present the same sets of controls for all the experiments.

We thank the reviewer for their comment. In general (and indeed all other data in Figure 3) we always compare age matched animals only, however in Figure 3i we present **both** that are age-matched and weight matched *mdx* animals. As *mdx* animals are generally heavier, and given we were observing weight gain over a period of time, we thought it pertinent to track the weight of similarly sized animals and included age matched animals for inclusivity.

May 18, 2021

Re: Life Science Alliance manuscript #LSA-2021-01014-TR

Dr. Corinne Anne Betts
University of Oxford
Paediatrics
Le Gros Clarke Building
South Parks Road
Oxford, Oxford OX1 3QX

Dear Dr. Betts,

Thank you for submitting your revised manuscript entitled "Dystrophins involvement in peripheral circadian SRF signalling" to Life Science Alliance. The manuscript has been seen by the original reviewers whose comments are appended below. While the reviewers continue to be overall positive about the work in terms of its suitability for Life Science Alliance, some important issues remain.

We apologize for this unusual and extended delay in getting back to you. As you will see from the comments below, while the reviewer agrees that the additional clarifications and explanations in the revised manuscript have improved the message, they have raised significant concerns about the molecular techniques used that must be addressed prior to further consideration of the manuscript.

- The reviewer is concerned about the data presentation of WBs in Figures 1 and 2. We agree that adding loading controls and comparing samples loaded side by side would be necessary for these.
- The reviewer also maintains the need for qPCR data for control gene loci not bound by SRF for the ChIP experiment in Fig 1G and supporting data to ensure that the myotubes are still alive and healthy after the different siRNA treatments
- About the concern raised by the reviewer about using beads-only control for the ChIP data in Fig 1G: Editorially, we revisited this matter given the reviewer's insistence on using IgG control. In our detailed assessment, we found that the suggested Thermofisher Dynabeads Protein A / G IP protocol (please see attached pdf) also points out that beads-only is an inappropriate control (Page 3, last paragraph). While we had mentioned that it would be okay to discuss this point, in light of the expert's opinion and looking at the company's recommendation, we suggest an IgG control be included - PLEASE DO REACH OUT TO ME IF YOU DISAGREE, HAPPY TO HAVE A FURTHER DISCUSSION ON THIS, OVER PHONE OR EMAIL.

Our general policy is that papers are considered through only one revision cycle; however, given that the suggested changes are of technical nature and required for ensuring the scientific rigor of the data provided, we are open to one additional short round of revision. Please note that I will expect to make a final decision without additional reviewer input upon resubmission.

Please submit the final revision within one month, along with a letter that includes a point by point response to the remaining reviewer comments.

B. MANUSCRIPT ORGANIZATION AND FORMATTING:

Sincerely,

Shachi Bhatt, Ph.D.
Executive Editor
Life Science Alliance
<http://www.lsjournal.org>
Tweet @SciBhatt @LSAJournal

Reviewer #1 (Comments to the Authors (Required)):

In this revised manuscript, the authors made many clarifications in the text that facilitates the comprehension for the reader. They also modified their summary figure which is now clearer. They better discuss the rationale for using both dKO and mdx mouse models, and they suggest possible explanations for the discrepancies observed between the models. Some major problems remain, mostly regarding some molecular biology techniques used in the study.

Major points:

1. When performing Western blot, samples that are going to be compared together must be loaded on the same gel, side by side. A loading control blot or Ponceau is also required, to ensure that

equal amounts of protein are loaded and transferred across all wells. Such minimal good laboratory practices are required for publication, in any journal. Please modify Figure 1A (load side by side and add loading controls), 1D (load side by side and add loading controls), 1E (load side by side), 2B (load side by side and add loading controls), 2C (load side by side). Figure 3A is done appropriately.

2. For ChIP experiment in Figure 1G, I appreciate that the authors presented the data as % input. I also asked that the authors use an IgG control instead of a beads-only control, and that they perform qPCR for control gene loci that should not be bound by SRF. This is, again, standard good laboratory practices.

3. In Figure 5, please add RT-qPCR data for control genes that stay stable in all siRNA conditions. That would ensure that the myotubes are still alive and healthy after the different siRNA treatments. Alternatively, the authors could provide data on myotubes viability after the different siRNA treatments.

4. It is very unfortunate that the authors used only one (1) commercially available control patient RNA for the gene expression analysis in Figure 1H, but at least it is now clearly indicated. Normal practice would be to include a minimum of three biological replicates.

Minor point:

The quantification of the F-G actin ratio suggests a 50% decrease. If such a drastic decrease is observed, this should be also visible by a simple phalloidin staining.

We would like to thank the reviewer for their time, positive feedback and suggestions to improve our manuscript entitled “*Dystrophin regulates peripheral circadian SRF signalling*”. We believe we have taken on-board all of the comments and addressed the questions raised by the Reviewer.

Reviewer #1

1. When performing Western blot, samples that are going to be compared together must be loaded on the same gel, side by side. A loading control blot or Ponceau is also required, to ensure that equal amounts of protein are loaded and transferred across all wells. Such minimal good laboratory practices are required for publication, in any journal. Please modify Figure 1A (load side by side and add loading controls), 1D (load side by side and add loading controls), 1E (load side by side), 2B (load side by side and add loading controls), 2C (load side by side). Figure 3A is done appropriately.

As previously mentioned, we now show all uncropped westerns blots in the ‘figure source data’ section, as set out in the journal guidelines: Western Fig 1, Western Fig 2, Western Fig 3, Western Fig S3. Providing uncropped versions and loading controls in the Figures would be difficult to visualise and cumbersome. The images in the ‘figure source data’ clearly show all samples were run on the same membrane (n= 3 or >), with associated loading controls. I have now referenced the relevant ‘figure source data’ files in the Figure legends so they may be more easily accessed.

2. For ChIP experiment in Figure 1G, I appreciate that the authors presented the data as % input. I also asked that the authors use an IgG control instead of a beads-only control, and that they perform qPCR for control gene loci that should not be bound by SRF. This is, again, standard good laboratory practices.

As discussed with Editor Eric Sawey, we have made the decision to remove the ChIP data.

3. In Figure 5, please add RT-qPCR data for control genes that stay stable in all siRNA conditions. That would ensure that the myotubes are still alive and healthy after the different siRNA treatments. Alternatively, the authors could provide data on myotubes viability after the different siRNA treatments.

As requested, we have run 2 other house-keeping genes to show that the myotubes were healthy at the time of collection. Both *Ywhaz* and *Atp5b* are stable at 100nM concentrations following *Srf*, *Acta* and *Dmd* knock-down when compared to non-targeting siRNAs.

Figure 1. qRT-PCR in H2KB4 myotubes following knock-down of *Srf*, *Acta* and *Dmd*

H2K 2B4 myotubes were transfected with 100nM siRNAs targeting the *Srf*, *Acta* and *Dmd* genes (*Srf* siRNA, blue, *Acta* siRNA grey and *Dmd* siRNA, red), and a non-targeting (NT siRNA, black) siRNA was used for control. Gene expression data indicates stable expression of house-keeping genes, *Ywhaz* and *Atp5b*. Data normalised to NT siRNA. Mean values reported with SEM.

4. It is very unfortunate that the authors used only one (1) commercially available control patient RNA for the gene expression analysis in Figure 1H, but at least it is now clearly indicated. Normal practice would be to include a minimum of three biological replicates.

The 'healthy control sample' was attained by pooling muscle biopsies from 2 healthy volunteers. We thank the Reviewer for understanding how difficult it was to attain these valuable samples.

August 2, 2021

RE: Life Science Alliance Manuscript #LSA-2021-01014-TRR

Dr. Corinne Anne Betts
University of Oxford
Paediatrics
Le Gros Clarke Building
South Parks Road
Oxford, Oxford OX1 3QX

Dear Dr. Betts,

Thank you for submitting your revised manuscript entitled "Dystrophins involvement in peripheral circadian SRF signalling". We would be happy to publish your paper in Life Science Alliance pending final revisions necessary to meet our formatting guidelines.

- please indicate the size next to each blot in the Figures
- please add the Twitter handle of your host institute/organization as well as your own or one of the first author in our system

LSA now encourages authors to provide a 30-60 second video where the study is briefly explained. We will use these videos on social media to promote the published paper and the presenting author. Corresponding or first-authors are welcome to submit the video. Please submit only one video per manuscript. The video can be emailed to contact@life-science-alliance.org

A. FINAL FILES:

B. MANUSCRIPT ORGANIZATION AND FORMATTING:

Sincerely,

August 4, 2021

RE: Life Science Alliance Manuscript #LSA-2021-01014-TRRR

Dr. Corinne Anne Betts
University of Oxford
Paediatrics
Le Gros Clarke Building
South Parks Road
Oxford, Oxford OX1 3QX
United Kingdom

Dear Dr. Betts,

Thank you for submitting your Research Article entitled "Dystrophins involvement in peripheral circadian SRF signalling". It is a pleasure to let you know that your manuscript is now accepted for publication in Life Science Alliance. Congratulations on this interesting work.

DISTRIBUTION OF MATERIALS:

Again, congratulations on a very nice paper. I hope you found the review process to be constructive and are pleased with how the manuscript was handled editorially. We look forward to future exciting submissions from your lab.

Sincerely,
